# A weakly solvating electrolyte towards practical rechargeable aqueous zinc-ion batteries

Xin Shi[1], Jinhao Xie[1], Jin Wang[1], Shilei Xie[2] ✉, Zujin Yang ᴼ[1] ✉ & Xihong Lu ᴼ[1] ✉

Structure deterioration and side reaction, which originated from the solvated $H_2O$, are the main constraints for the practical deployment of both cathode and anode in aqueous Zn-ion batteries. Here we formulate a weakly solvating electrolyte to reduce the solvating power of $H_2O$ and strengthen the coordination competitiveness of $SO_4^{2-}$ to $Zn^{2+}$ over $H_2O$. Experiment results and theoretical simulations demonstrate that the water-poor solvation structure of $Zn^{2+}$ is achieved, which can (i) substantially eliminate solvated-$H_2O$-mediated undesirable side reactions on the Zn anode. (ii) boost the desolvation kinetics of $Zn^{2+}$ and suppress Zn dendrite growth as well as structure aberration of the cathode. Remarkably, the synergy of these two factors enables long-life full cells including $Zn/NaV_3O_8 \cdot 1.5H_2O$, $Zn/MnO_2$ and $Zn/CoFe(CN)_6$ cells. More importantly, practical rechargeable AA-type Zn/NVO cells are assembled, which present a capacity of 101.7 mAh and stability of 96.1% capacity retention after 30 cycles at 0.66 C.

Rechargeable aqueous Zn-ion batteries (AZIBs) are promising electrochemical devices for stationary energy storage that have been widely investigated by both academia and industry because of the intrinsic merits of Zn anode such as high capacity (820 mAh g⁻¹), low redox potential (− 0.762 V vs. the standard hydrogen electrode (SHE)) and relatively low cost ($1.45/lb)[1–8]. When coupling the Zn anode with an appropriate cathode (Mn-based, V-based, Prussian blue analogs-based, etc.) and electrolyte ($Zn(OTf)_2$, $ZnSO_4$, etc.), AZIBs can achieve satisfying energy and power density comparable or even superior to commercial aqueous devices like Ni-Zn batteries and lead-acid batteries[9–14]. In the last decade, substantial progress have been made in the development of high-performance AZIBs. Nevertheless, the limited cycling durability, initiated by dendrite growth and side reactions of Zn anode and structure degradation of the cathode during repeated charge/discharge courses, remains the major obstacle that plagues their practical deployment[15–17]. The electrode failure of AZIBs is highly dependent on the interfacial structures between electrode and electrolyte, also known as the electric double layer (EDL)[18]. On the

basis of the Bockris–Devanathan–Müller model (Supplementary Fig. 1), the key issues faced by Zn anode and cathode materials can be well explained: (i) Zn dendrite growth is caused by the uneven $Zn^{2+}$ flux from outer Helmholtz layer to Zn anode, which is associated with the strong coordination bond between $Zn^{2+}$ and solvated $H_2O$ in $[Zn(H_2O)_6]^{2+}$ that leads to sluggish desolvation kinetics of $Zn^{2+}$. Such poor desolvation kinetics can also cause the co-insertion of $Zn^{2+}$-$H_2O$ complex into the cathode host, leading to irreversible lattice expansion and ultimately structure degradation[19]. (ii) During the desolvation process of $[Zn(H_2O)_6]^{2+}$, a large number of active $H_2O$ molecules are released and in contact with Zn anode, which in turn triggers hydrogen evolution reaction (HER), corrosion and other side reactions[20]. Therefore, decreasing the number of solvated $H_2O$ is the key to inhibiting the failure of both cathode and anode in AZIBs.

To date, some electrolyte modification strategies such as adding high-concentration electrolyte salts and introducing high donor number (DN) organic additives have been proposed to minimize the solvated $H_2O$. Concretely, 1 m $Zn(TFSI)_2$ + 20 m LiTFSI highly

[1]MOE of the Key Laboratory of Bioinorganic and Synthetic Chemistry, The Key Lab of Low-carbon Chem & Energy Conservation of Guangdong Province, School of Chemistry, School of Chemical Engineering and Technology, Sun Yat-Sen University, Guangzhou, PR China. [2]School of Environment and Civil Engineering, Guangdong Engineering and Technology Research Center for Advanced Nanomaterials, Dongguan University of Technology, Dongguan, PR China. ✉e-mail: xieshil@dgut.edu.cn; yangzj3@mail.sysu.edu.cn; luxh6@mail.sysu.edu.cn

concentrated electrolyte (HCE) was proposed to suppress the content of solvated $H_2O$ and enable a $Zn/O_2$ cell with extended life beyond 200 cycles[21]. Despite the inspiring achievements, the viscosity of HCE is too thick, resulting in uneven diffusion of $Zn^{2+}$ and poor rate capability of full cells. Alternatively, the solvated $H_2O$ can be partially replaced by introducing organic molecules with DN greater than $H_2O$ such as dimethyl sulfoxide[22], triethyl phosphate (TEP)[23], ethylene glycol (EG)[24], polyethylene glycol[19], acetamide[25]. For example, it is reported that the addition of 68% v/v EG to 3 M $ZnSO_4$ could extend the cycling durability of $Zn/Zn_xV_2O_5 \cdot nH_2O$ cells to more than 500 cycles at $0.5\,A\,g^{-1}$, seeing a nearly 20-time lifespan enhancement in contrast to bare 3 M $ZnSO_4$[24]. A similar life-extending effect was also observed by using triethyl phosphate (TEP) as the additive[23]. However, these organic additives are used to replace the solvated $H_2O$ due to their stronger interaction with $Zn^{2+}$, and such interaction will further slowdown the desolvation kinetics of $Zn^{2+}$, which is not conducive to the even diffusion of $Zn^{2+}$ and may even aggravate the growth of Zn dendrites as well as structure fading of cathode materials at high current densities. Therefore, strategic minimization of solvated $H_2O$ to simultaneously stabilize anode and cathode of AZIBs still remains challenging.

A more basic way to reduce the number of solvated $H_2O$ is by weakening the intrinsic solvating ability of $H_2O$[26]. The main factors that evaluate the solvating ability of the solvent are the dielectric constant (ε) and DN, ε describes the electrostatic force between the solvated ions and DN influences the nucleophilic behavior of solvent[27]. Considering $H_2O$ with large ε (80.4) and DN (18) is the predominant solvent in AZIBs and exhibits a high solvating ability for zinc salts, it is hypothesized the solvating power of $H_2O$ can be limited by additives with low ε and DN. Herein, we formulate a weakly solvating electrolyte (WSE) by introducing low ε and DN butanone as an additive to 0.5 M $ZnSO_4$ to build long-life AZIBs. Experiments and theoretical simulation indicate that WSE can (i) reduce the number of solvated $H_2O$ and inhibit side reactions on Zn anode. (ii) boost the desolvation kinetics of $Zn^{2+}$ and suppress dendrite growth as well as structure fading of cathode materials. Since $H_2O$ and anions are competing to enter the solvation sheath of $Zn^{2+}$, reducing the number of solvated $H_2O$ can enable more anions to enter the solvation sheath of $Zn^{2+}$, forming a water-poor solvation structure of $[(Zn^{2+})(H_2O)_{4.3}(SO_4^{2-})_{1.3}(C_4H_8O)_{0.4}]$. During the $Zn^{2+}$ deposition process on Zn anode or the $Zn^{2+}$ insertion process on cathode, the negatively charged electrode surface has electrostatic repulsion with the anion in the solvation sheath of $Zn^{2+}$, which helps to accelerate the desolvation kinetics of $Zn^{2+}$ and promote the uniform diffusion of $Zn^{2+}$ (Fig. 1a). Consequently, Zn anode cycling in WSE exhibits outstanding stability at an incredibly high current density, areal capacity and depth of discharge ($60\,mA\,cm^{-2}$, $60\,mAh\,cm^{-2}$ and 88.2%). Three representative AZIBs including $Zn/NaV_3O_8 \cdot 1.5H_2O$ (NVO), $Zn/MnO_2$, and $Zn/CoFe(CN)_6$ cells using WSE were assembled and show excellent cycling stability. More encouragingly, rechargeable AA type Zn/NVO AZIBs were fabricated, which can deliver a high capacity of 101.7 mAh and outstanding stability of 96.1% capacity retention after 30 cycles at 0.66 C.

## Results

### Solvation structure of $Zn^{2+}$ in WSE

WSE consists of water, co-solvent (additive), and electrolyte salt. Theoretically, the construction of a weakly solvated environment requires the co-solvent to meet the following conditions: (1) miscible with water; (2) low polarity (i.e., small ε); and (3) low nucleophilic ability to metal ions (i.e., low DN). Supplementary Table S1 lists common organic solvents that meet both ε and DN values less than water. However, the smaller ε value indicates that the solvent is more non-polarized and difficult to dissolve in strong polarized water. By analyzing whether different solvents can generate hydrogen bonds with water to reduce the Gibbs free energy of the

system, we selected butanone as the main co-solvent for this study due to its good solubility in water. To investigate the suitable dosage of butanone, we prepared different concentrated $ZnSO_4$ electrolytes (0.3–2 M) and their corresponding butanone-saturated electrolytes in the preliminary experiment. By testing the cycling stability of Zn/NVO full cells with different electrolytes, the optimized electrolyte formula was determined. As shown in Supplementary Fig. 2, Zn/NVO full cell shows the best cycling performance (∼97.8% capacity retention after 1000 cycles). Therefore, 0.5 M aqueous $ZnSO_4$ solution with saturated butanone additives (12.5% v/v) is proposed for high-performance AZIBs assembly. The low-concentration electrolyte (LCE, 0.5 M $ZnSO_4/H_2O$) without additive is the base electrolyte. As revealed by the Raman spectra in Fig. 1b, the signal corresponding to $v(SO_4^{2-})$ vibration shifts to a higher wavenumber after the addition of butanone (from 977.1 to $977.3\,cm^{-1}$ in WSE). Moreover, an obvious difference at a range of $985.8$-$1008.9\,cm^{-1}$ can be observed when overlapping the Raman spectra of LCE and WSE (Supplementary Fig. 3). Such variation trend indicates the solvation structure of $Zn^{2+}$ transforms from pure solvent-separated ion pair (SSIP) to the co-existence of SSIP and contact ion pair (CIP)[28]. Meanwhile, the $v(H_2O)$ vibration at $3235$–$3245\,cm^{-1}$ also experiences weakly positive shift in WSE, signifying the existence of strong intermolecular hydrogen bonds between butanone and $H_2O$ (Supplementary Fig. 4). The solvation structure variation of $Zn^{2+}$ in WSE are also testified by the butanone-induced positive shift of $v(SO_4^{2-})$ band in Fourier transform infrared spectroscopy (FT-IR) (Fig. 1c). [1]H nuclear magnetic resonance (NMR) spectra further verify the strong hydrogen bonding effect between butanone and $H_2O$. The [1]H signal shifts to lower ppm after the introduction of butanone (Supplementary Fig. 5), e. g. from 4.70 to 4.68 ppm for WSE. This is because the electron density of hydroxyl groups is increased by the electron donor carbonyl groups due to the formation of hydrogen bonds (HBs). The ionic conductivity of different electrolytes is displayed in Supplementary Fig. 6. We can see the ionic conductivity of aqueous $ZnSO_4$ decreases with the addition of butanone, which may be caused by the reduced anion mobility due to the introduction of butanone that shows hydrogen bonding effect with $SO_4^{2-}$. Nevertheless, the existence of electronegative $SO_4^{2-}$ in PSS can greatly decrease the electrostatic potential of $[Zn(H_2O)_x]^{2+} \cdot [OSO_3^{2-}]_y$, thus facilitating its diffusion. $Zn^{2+}$ diffusion coefficient calculated from Molecular dynamics (MD) methods also well supports this point (Supplementary Fig. 7). Accordingly, the $Zn^{2+}$ transference number ($t_+$) of WSE increases from ∼0.34 to ∼0.44 (Supplementary Fig. 8). Moreover, the freezing point of WSE is decreased to −25.8 °C, indicating the enhanced anti-freezing property of WSE (Supplementary Fig. 9). The pH values of LCE and WSE are 4.28 and 4.02. Since butanone does not ionize in aqueous solution, butanone itself has no impact on the pH variation of WSE. However, it cannot dissolve $ZnSO_4$ and the introduction of butanone decreases the content of $H_2O$ from 40 mL in LCE to 35 mL in WSE. Therefore, the $H_2O$ in WSE would dissolve more $ZnSO_4$ salt compared to that in LCE. Considering $ZnSO_4$ is hydrolyzing in aqueous solution, more $ZnSO_4$ dissolved by $H_2O$ in WSE will lead to more severe hydrolysis thus the pH of WSE is slightly lower than that of LCE. The viscosity of LCE and WSE are 1.25 and $1.68\,mP\,s^{-1}$, respectively.

MD methods were further applied to simulate the solvation structure disparities of LCE and WSE. As displayed in Fig. 1d and Supplementary Fig. 10, the PSS of $Zn^{2+}$ only consists of $H_2O$ in LCE while partial $H_2O$ is replaced by $SO_4^{2-}$ after the introduction of butanone. The average coordination number (ACN) and radial distribution functions (RDFs) of the two systems are shown in Fig. 1e, f and Supplementary Fig. 11. Apparently, no anion is found in the PSS of $Zn^{2+}$ in LCE systems. In contrast, in butanone-containing 0.5 M $ZnSO_4$, the ACN of $SO_4^{2-}$ in the PSS accounts for 1.3, associated with the ACN decrease of $H_2O$ from

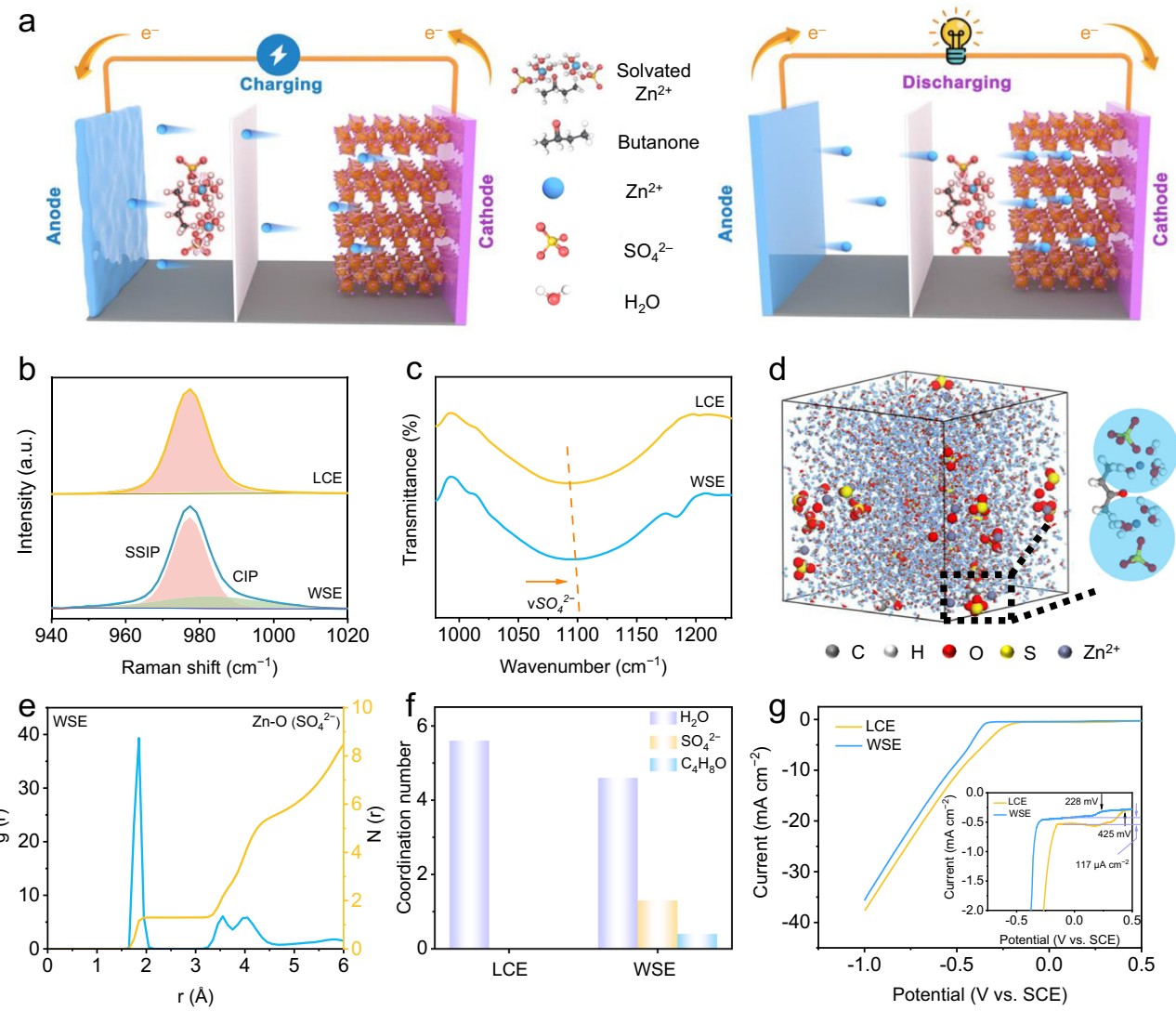

**Fig. 1 | Solvation structure of Zn²⁺ in LCE and WSE.** Schematic illustrations of the working mechanism of **a** WSE. **b** Raman spectra, **c** FT-IR spectra of LCE and WSE. **d** 3D snapshot of WSE obtained from MD simulations and partially enlarged snapshot representing Zn²⁺ solvation structure. **e** RDFs for Zn²⁺-OSO₃²⁻ collected from MD simulations in WSE. **f** Coordination number of Zn²⁺ with different species in LCE and WSE. **g** LSV curves in LCE and WSE at 5 mV s⁻¹. The inset is the enlarged LSV curves in LCE and WSE.

5.6 to 4.3, realizing the successful formation of WSE. The partial replacement of $H_2O$ by $SO_4^{2-}$ ions should be attributed to the lower solvating ability of solvent in WSE. The impact of the dielectric constant on the Zn²⁺-solvent interactions can be depicted by classical physiochemical laws:

$$U_{Zn^{2+}-solvent} = \frac{-1}{4\pi\varepsilon} \times \frac{q\mu\cos\theta}{r^2} \quad (1)$$

where $U$ is the interaction of Zn²⁺-solvent, $\varepsilon$ is the dielectric constant, $q$ is the charge of ion, $\mu$ is the dipole moment of dipole, $r$ is the distance between ion and the center of dipole, and $\theta$ is the dipole angle relative to the line $r$ joining the ion and the center of the dipole. In WSE, butanone strongly reduces the $\varepsilon$ of solvent and lowers the interaction between Zn²⁺ and solvent. As the entrance ticket to PSS of Zn²⁺ is competed by solvents and anions, decreased interaction between Zn²⁺ and solvent means anion is possible to enter PSS of Zn²⁺ thus forming $[(Zn^{2+})(H_2O)_{4.3}(SO_4^{2-})_{1.3}(C_4H_8O)_{0.4}]$ CIP. The Zn²⁺ plating and stripping potential was tested on the Ti electrode. As shown in Fig. 1g, WSE can postpone the onset potential of HER from 425 mV to 228 mV vs. SCE. Besides, the current of HER is eliminated by 117 μA cm⁻² with the

addition of butanone, indicating the changed solvation structure can highly suppress HER activity in WSE.

## Stripping/plating reversibility of Zn anode in WSE

To assess the impact of WSE on the plating/stripping courses of Zn anode, Zn/Zn symmetric cells using LCE and WSE were assembled and tested. cells using WSE exhibit improved cycling lifespan over that with LCE at 1 mA/1 mAh cm⁻² and 5 mA/5 mAh cm⁻² (Supplementary Fig. 12). More Encouragingly, cells using WSE can still deliver stability for more than 70 h at high current density (60 mA cm⁻²) and capacity (60 mAh cm⁻²) with a high DOD of 88.2% while the cells using LCE can not work (Fig. 2a). To the best of our knowledge, such performance far more exceeds the state-of-art Zn anode (Supplementary Table S2). The plating/stripping test was further carried out in Ti/Zn asymmetric cells based on LCE and WSE. As shown in Fig. 2b and Supplementary Fig. 13, after the relatively low CE in the first few cycles which is owing to the lattice fitting, the Ti/Zn cells adopting WSE can deliver a high CE of ~99.9% as well as long-term stability (>300 h). In opposite, the Ti/Zn cells using LCE are trapped with short cycling life (<100 h) and low CE (<90%), which could be attributed to the undesirable dendrite growth and HER.

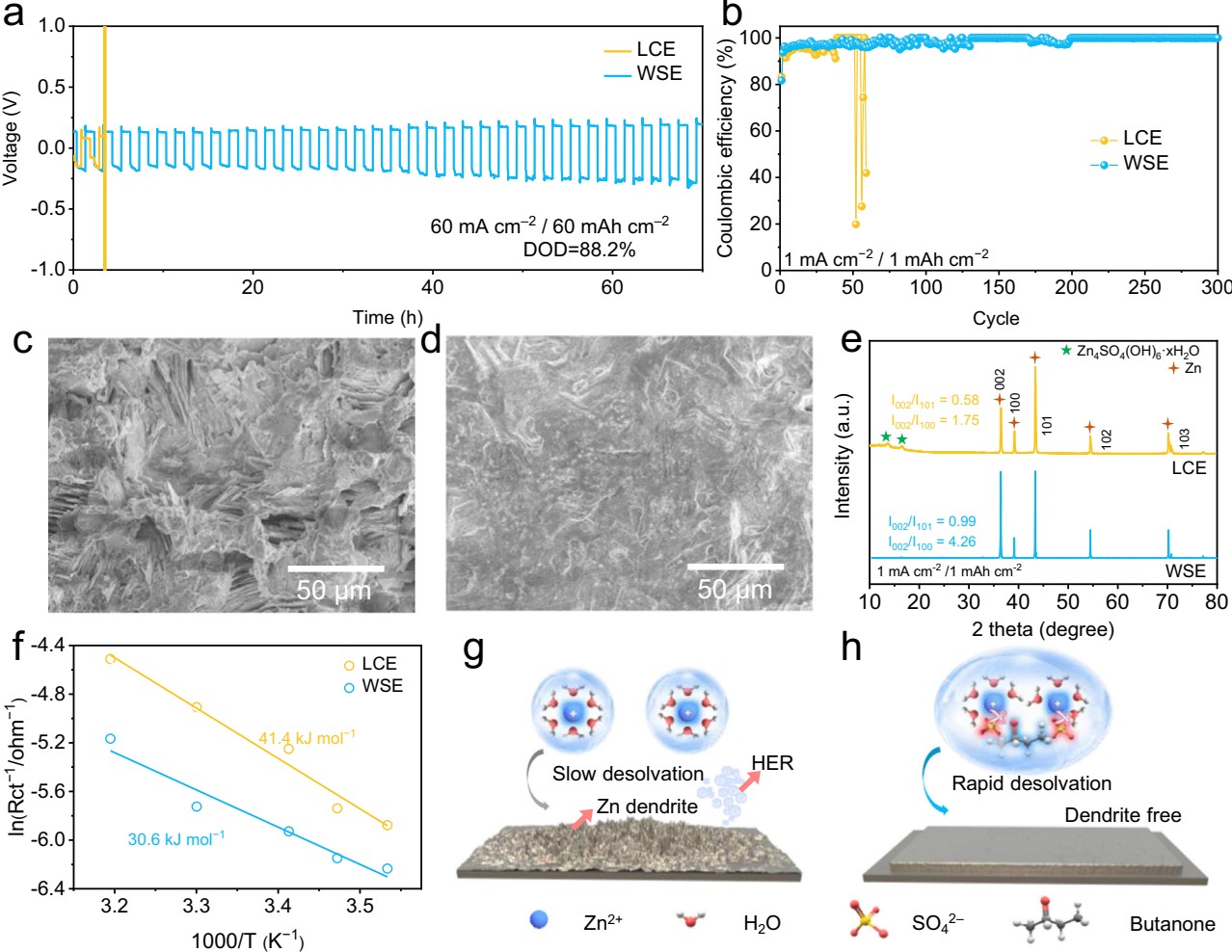

**Fig. 2 | Electrochemical performance of Zn anode in WSE and corresponding protection mechanism study. a** Galvanostatic cycling profiles of symmetric cells with LCE and WSE at 60 mA/60 mAh cm$^{-2}$. **b** Coulombic efficiency (CE) measurements of Ti/Zn cells under different electrolyte systems at 1 mA/1 mAh cm$^{-2}$. SEM images of Zn foil after 50 cycles at 1 mA/1 mAh cm$^{-2}$ in **c** LCE and **d** WSE. **e** XRD pattern of Zn foil cycled after 50 cycles at 1 mA/1 mAh cm$^{-2}$ in LCE and WSE. **f** Plotted charge transfer resistance of Zn/electrolyte interface measured from Zn/Zn symmetrical cell using LCE and WSE. **g** Schematic diagram of how Zn anode works in **g** LCE and **h** WSE.

## Protection mechenism of WSE for Zn anode

The reason for the enhanced stability of Zn anode in WSE is monitored via SEM, XRD and electrochemical measurements. SEM presents the rough and porous surfaces of Zn anode with irregular Zn dendrite in LCE at 1 mA/1 mAh cm$^{-2}$ after 50 cycles (Fig. 2c), which deteriorate as the current densities and capacities increased to 10 mA/10 mAh cm$^{-2}$ (Supplementary Fig. 14a). In contrary, flat surfaces constituted with densely deposited Zn can be observed at all test conditions in WSE (Fig. 2d and Supplementary Fig. 14b). XRD characterizations were carried out to evaluate the crystal structure of Zn deposits after 50 plating/stripping cycles at 1 mA/1 mAh cm$^{-2}$. As illustrated in Fig. 2e, the [002] peak of Zn deposits is the strongest in WSE. The intensity ratios of [002] peak to [101] peak are respectively 0.44 and 0.99 for LCE and WSE, suggesting the increased [002] planes for Zn deposits in WSE. Considering the HCP lattice (i.e., space group: P63/mmc) of Zn metal, it is speculated that the deposited Zn are piled up perpendicular to the c axis, thus leading to a smooth surface as observed from SEM images[29]. When the cycling parameter is changed to 10 mA/10 mAh cm$^{-2}$, the intensity ratio of [002] peak to [101] peak of Zn in WSE increases to 2.91, indicating more favorable dense deposit of Zn at high rate (Supplementary Fig. 15). In addition, two pronounced peaks at 13.7° and 16.7° can be found for Zn plates cycled in LCE, which is due to the HER and the formation of

Zn$_4$SO$_4$(OH)$_6$·4H$_2$O by-products in ZnSO$_4$ electrolyte[30]. On the contrary, no characteristic peak of Zn$_4$SO$_4$(OH)$_6$·4H$_2$O can be found for Zn electrode in WSE, suggesting HER is highly suppressed. The increased corrosion potential further confirms the enhanced anticorrosion ability of WSE for Zn anode protection (Supplementary Fig. 16).

The Zn$^{2+}$ deposition process was further understood via electrochemical analysis techniques. The activation energy ($E_a$) for the desolvation kinetics of Zn$^{2+}$ ions based on Arrhenius equation in different electrolytes was compared in Fig. 2f. The $E_a$ values are 41.4 and 30.6 kJ mol$^{-1}$ in LCE and WSE electrolytes, respectively, indicating the lower desolvation energy barrier in WSE, which is due to the unique anion type solvation structure of Zn$^{2+}$ in WSE that facilitates its desolvation process via the electrostatic repulsion effect with negatively polarized Zn anode[31]. Thanks to the improved desolvation kinetics of Zn$^{2+}$ in WSE, the further nucleation step of Zn is also boosted. As presented in Supplementary Fig. 18, the nucleation overpotential of Zn electrode in WSE is lower than that in LCE across a range of current densities (1–10 mA cm$^{-2}$), which is beneficial for the uniform growth of Zn deposits. Therefore, the enhanced stability of Zn anode in WSE can be attributed to: (i) a water-poor Zn$^{2+}$ solvation structure is formed in WSE which suppresses HER corrosion and other side reaction; (ii) such solvation structure of Zn$^{2+}$ is beneficial to fast Zn$^{2+}$ migration and

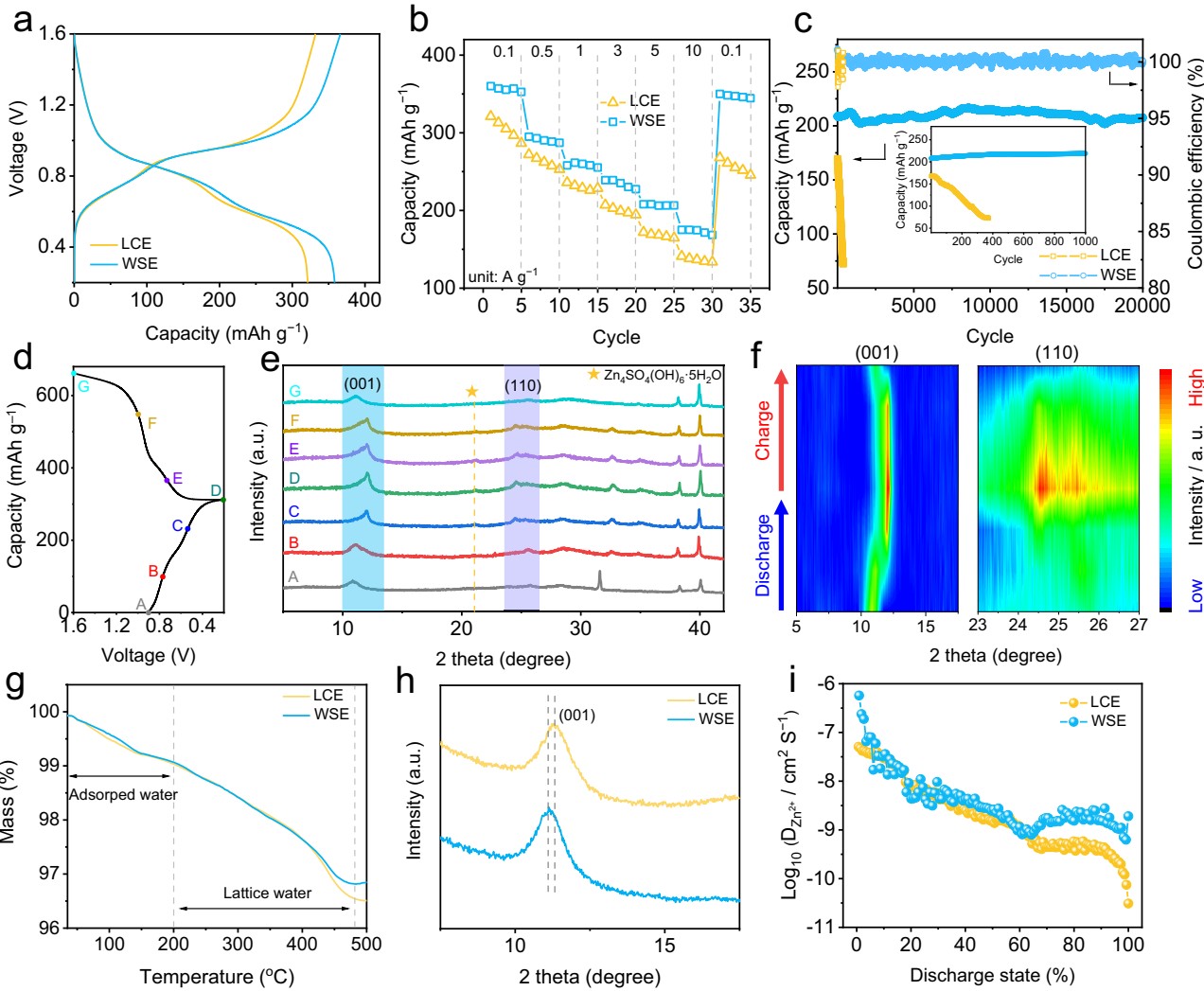

**Fig. 3 | Electrochemical performance and structure evolution of NVO cathode in WSE. a** GCD curves at 0.1 A g⁻¹, **b** rate capability comparison and **c** cycling stability comparison at 5 A g⁻¹ of Zn/NVO cells in LCE and WSE. The inset in Fig. 3c is the enlarged cycling stability at 5 A g⁻¹ of Zn/NVO cells in LCE and WSE. **d** GCD curves of NVO electrode using WSE at 0.1 A g⁻¹, **e** ex-situ XRD patterns and **f** corresponding contour map. **g** TG curves and **h** XRD patterns of NVO electrode after 50 cycles at 0.1 A g⁻¹ in LCE and WSE. **i** Diffusivity coefficient at various states of discharge in LCE and WSE according to GITT measurement.

desolvation process, which further contribute to the uniform nucleation and growth (Fig. 2g).

**Robust cycling stability of cathode materials in WSE**

To examine whether WSE affects the cycling sustainability of cathode, full cells were assembled by using NVO as the object of study. Typical scanning electron microscopy (SEM) and X-ray diffraction (XRD) characterization results of NVO can be found in Supplementary Fig. 19. The cyclic voltammetry (CV) curves at initial three cycles of Zn/NVO cell with WSE shows two pairs of redox peaks attributed to V$^{5+}$↔V$^{4+}$ and V$^{4+}$↔V$^{3+}$ (Supplementary Fig. 20). Note CV curves at the 2$^{nd}$ and 3$^{rd}$ cycles are almost coincident, indicating its good electrochemical stability. Fig. 3a shows galvanostatic charge–discharge (GCD) curves at 0.1 A g⁻¹ of Zn/NVO cells with LCE and WSE, respectively. The capacity superiority of Zn/NVO cells follows the order of WSE and LCE, indicating the addition of butanone can substantially promote the Zn²⁺ storage ability of NVO. The positive effect of butanone additive is also proved by the rate capability comparison, in which Zn/NVO cell with WSE shows the best capacity retention (48.5%) from 0.1 to 10 A g⁻¹ (Fig. 3b and Supplementary Fig. 21). The existence of butanone can also effectively stabilize Zn/NVO cells during cycling. In WSE, the device maintains a high capacity of 313.7 mAh g⁻¹ after 100 cycles at a

moderate rate of 0.1 A g⁻¹, way better than the counterparts with LCE (127.7 mAh g⁻¹, Supplementary Fig. 22). The increase of the test current density to 5 A g⁻¹ further widens the performance gap of these two devices. Zn/NVO cell with WSE exhibits a capacity retention of 99.1% after 20000 cycles while that with LCE fails quickly after only 400 cycles (Fig. 3c). To highlight the effectiveness of WSE, we compare the cycling stability of Zn/NVO cells with other recently reported works focusing on electrolyte design. As shown in Supplementary Table S3, the capacity retention of Zn/NVO cells using WSE is higher than that of Zn/V cells based on other electrolyte chemistry, confirming the effectiveness of WSE. Benefitted from the improved anti-freezing ability of WSE, the Zn/NVO cells with WSE can maintain a capacity of 178.7 mAh g⁻¹ after 100 cycles with a capacity retention of 66.4% at −20 °C (Supplementary Fig. 23), while its counterpart can barely work. Additionally, the stability of Zn/NVO cells in WSE with a high mass loading cathode (~10 mg cm⁻², denoted as Zn/h-NVO) was investigated. A high capacity of 325.1 mAh g⁻¹ (3.25 mAh cm⁻²), a high capacity retention of 84.1% and a remarkable capacity of 270.2 mAh g⁻¹ after 100 cycles at 0.1 A g⁻¹ can be achieved for Zn/h-NVO, indicating the validity of WSE for stabilizing high-capacity V-based AZIBs (Supplementary Fig. 24). More encouragingly, we find that the usage of WSE can also strikingly improve the rate capability and cycling stability of other

AZIBs systems, e. g. Zn/CoFe(CN)$_6$ cells and Zn/MnO$_2$ cells (Supplementary Figs. 25 and 26), proving the universality of the as-proposed WSE strategy. In addition, the cycling stability of Zn/CoFe(CN)$_6$ and Zn/MnO$_2$ cells with high cathodic mass loading is presented in Supplementary Fig. 27. Compared to cells using LCE, extended capacity retention of 74.4% and 57.9% can be obtained for Zn/CoFe(CN)$_6$ and Zn/MnO$_2$ cells using WSE. The Ragone plots of Zn/NVO, Zn/CoFe(CN)$_6$ and Zn/MnO$_2$ cells using WSE are shown in Supplementary Fig. 28 (based on the mass loading of cathode materials), in which energy densities of 269, 170.5, and 360.2 Wh kg$^{-1}$ are obtained for Zn/NVO, Zn/CoFe(CN)$_6$ and Zn/MnO$_2$ cells.

### Protection mechanism of WSE for cathode

To investigate the underlying reason for the differentiated electrochemical performance of Zn/NVO cells in different electrolytes, several ex-situ characterizations were conducted to monitor the structure and valence change of NVO cathode during repeated charge/discharge processes. Fig. 3e, f displayed the ex-situ XRD patterns of NVO in WSE at representative discharge (A-D) and charge (D-G) states (Fig. 3d). Note that the reversible appearance (discharge to D) and disappearance (charge to G) of the peak at 21.1° can be assigned to Zn$_4$SO$_4$(OH)$_6$·4H$_2$O, suggesting the existence of reversible H$^+$ insertion/extraction behaviors in the NVO electrode[32]. Zn 2$p$ XPS spectra reveal the reversible Zn$^{2+}$ insertion/extraction mechanism of NVO electrode, in which the strong intensity of Zn 2$p$ spectra at state D indicates the insertion of Zn$^{2+}$ while it becomes weaker at state G (Supplementary Fig. 29). The positive shift of Zn 2p spectra is due to the formation of Zn$_4$SO$_4$(OH)$_6$·4H$_2$O at state D, in good agreement with the XRD results. In addition, the (001) plane gradually shifts from 11.0° to 12.1° at state D and then recover to its initial 2 theta degree upon state G, suggesting the decreased d-space of (001) plane at state D. Conversely, the negative shift of (110) plane can be observed at state D, indicating the increased d-space of (110) plane (Fig. 3f). Such phenomenon is due to the polarity of inserted H$^+$, Zn$^{2+}$ and their hydrated ions that cause structure aberration. To highlight the difference between NVO cycled in LCE and WSE, we dissected the Zn/NVO cells with LCE and WSE after 50 cycles at a low current density of 0.1 A g$^{-1}$. We set the cutoff voltage of NVO with both electrolytes as 1.6 V (fully charged state) to investigate the inserted lattice H$_2$O into NVO in different electrolytes. Several characterizations including thermogravimetry (TG), XPS, XRD and SEM were performed to evaluate the structure evolution of different NVO electrodes. In TG curves, the weight loss above 200 °C could be attributed to the loss of lattice H$_2$O. Fig. 3g suggests the NVO cycled in WSE shows less loss of lattice H$_2$O, indicating the amount of co-inserted H$_2$O is decreased to some extent. O 1$s$ XPS spectra further confirm the decreased H$_2$O of NVO with WSE, as the intensity of peak representing O-H at 532.5 eV of NVO with WSE is lower than that with LCE (Supplementary Fig. 30). As shown in Fig. 3h, the XRD of NVO cycled in LCE experiences a more obvious lattice expansion than that in WSE (11.3° to 11.1°), as revealed by the more deviated (001) plane from the pristine NVO, suggesting the mitigatory lattice variation of NVO in WSE. Such lattice expansion of NVO in LCE may cause strain and the consequent crack formation, as revealed by SEM images (Supplementary Fig. 31). Much more cracks were formed on the NVO electrode cycled in LCE while a flat surface is observed for NVO cycled in WSE, indicating the protective ability of WSE for NVO at the electrode level. Based on the above experimental results, it can be concluded that H$_2$O co-insertion happens in both LCE and WSE, except more H$_2$O is co-inserted into NVO in LCE during the cycling process. Moreover, the insertion of a suitable amount of H$_2$O is beneficial for the energy storage process of NVO while excess insertion of H$_2$O can cause capacity fading. In addition to the structure deformation, the cathode dissolution should also be paid attention for the capacity decay of NVO. NVO electrodes were first immersed in LCE and WSE for 7 days to test their solubility. As shown in Supplementary

Fig. 32, the bottle containing WSE shows a lighter yellow compared with the LCE, indicating the dissolution of NVO is diminished by butanone. We further performed the dissolution test in 2032-type coin cells, in which 80 μL electrolytes were used. After resting for 7 days, the separators of cells with LCE and WSE were taken out and further used for the inductively coupled plasma (ICP) test. As shown in Supplementary Fig. 33, the atom ratio of V element for LCE and WSE is 0.082 mg L$^{-1}$ and 0.01 mg L$^{-1}$, indicating that WSE can highly suppress the dissolution of NVO under the actual working condition. XRD patterns of NVO soaked in LCE and WSE for 7 days are shown in Supplementary Fig. 34. The (001) and (110) planes of NVO soaked in both LCE and WSE show no deviate from the pristine NVO except the (001) plane of NVO soaked in LCE is missing, indicating the structure of NVO is destroyed in LCE due to the dissolution issue. To highlight the negative impact of cathode dissolution on the overall performance of Zn/ZVO, NVO electrodes that underwent the dissolution tests (resting the Zn/NVO cells with LCE and WSE for 7 days) were subjected to GCD tests. Compared to the cells that did not experience the dissolution test (Fig. 3a), the NVO electrodes tested in LCE and WSE show capacity loss of 44.4 % and 7.8 % in the first cycle, respectively, indicating the WSE can also suppress the cathode dissolution of NVO (Supplementary Fig. 35).

The reaction kinetics of NVO in different electrolytes were instigated by EIS, CV and galvanostatic intermittent titration technique (GITT). The EIS spectra of Zn/NVO full cell with LCE and WSE after activation are shown in Supplementary Fig. 36. Zn/NVO full cell with WSE shows smaller Warburg resistance (R$_w$) than that in LCE, indicating the facilitated ion transfer process for Zn$^{2+}$ ion storage due to the changed solvation structure of Zn$^{2+}$. CV curves of the Zn/NVO cells are tested at various scan rates (0.2-1 mV s$^{-1}$, Supplementary Fig. 37). The b values for NVO in WSE are 0.576, 0.609, 0.636, and 0.624, implying the charge storage process is synergistically controlled via a combination of ionic diffusion and capacitance behaviors. Note both b values and capacitance contribution at all scan rates of Zn/NVO cells using WSE are higher than that in pure ZnSO$_4$ electrolytes, suggesting the faster reaction kinetics of NVO in WSE. GITT measurement was performed to evaluate the ion-diffusion coefficient (D$_{ion}$) in NVO cathode (Fig. 3i and Supplementary Fig. 38). The ion diffusion coefficient (D$_{ion}$) at low DOD reflects the H$^+$ insertion kinetics while D$_{ion}$ at high DOD represents the Zn$^{2+}$ insertion kinetics. The D$_{ion}$ of H$^+$ insertion is similar in LCE and WSE, indicating the introduction of butanone has a negligible effect on H$^+$ insertion kinetics. However, the D$_{ion}$ of Zn$^{2+}$ insertion in WSE is higher than that in LCE, suggesting the faster Zn$^{2+}$ insertion kinetics with newly recruited butanone in WSE. This phenomenon is because that (i) WSE shows boosted Zn$^{2+}$ desolvation kinetics; (ii) the overdosed co-inserted H$_2$O is minished in WSE. The excess insertion of H$_2$O in LCE can block the Zn$^{2+}$ insertion pathway, thus increasing the insertion resistance[33]. On the opposite, the transformed solvation structure of Zn$^{2+}$ with butanone additive can facilitate the desolvation process of Zn$^{2+}$ and decrease the amount of co-inserted H$_2$O. Therefore, the protection mechanism of WSE for NVO can be concluded as: (i) the excess co-insertion of H$_2$O is suppressed and the structure aberration/collapse of NVO during cycling is relieved; (ii) the dissolution of NVO is greatly mitigated because the activity of H$_2$O is reduced owing to the intense hydrogen bonding interaction between H$_2$O and butanone.

### Fabrication of AA-type Zn/NVO cell based on WSE

AA-type batteries are used in a wide range of scenarios in daily life, which request battery systems with high safety, performance, and sustainability. Therefore, to further testify the feasibility of WSE for practical application, rechargeable AA type Zn/NVO cells (denoted as AA-Zn/NVO) based on WSE were assembled (Fig. 4a). This cell prototype is realized by rolling 4 × 23 cm$^2$ NVO electrode, brown paper separator and Zn foil to tight and fill them into stainless steel

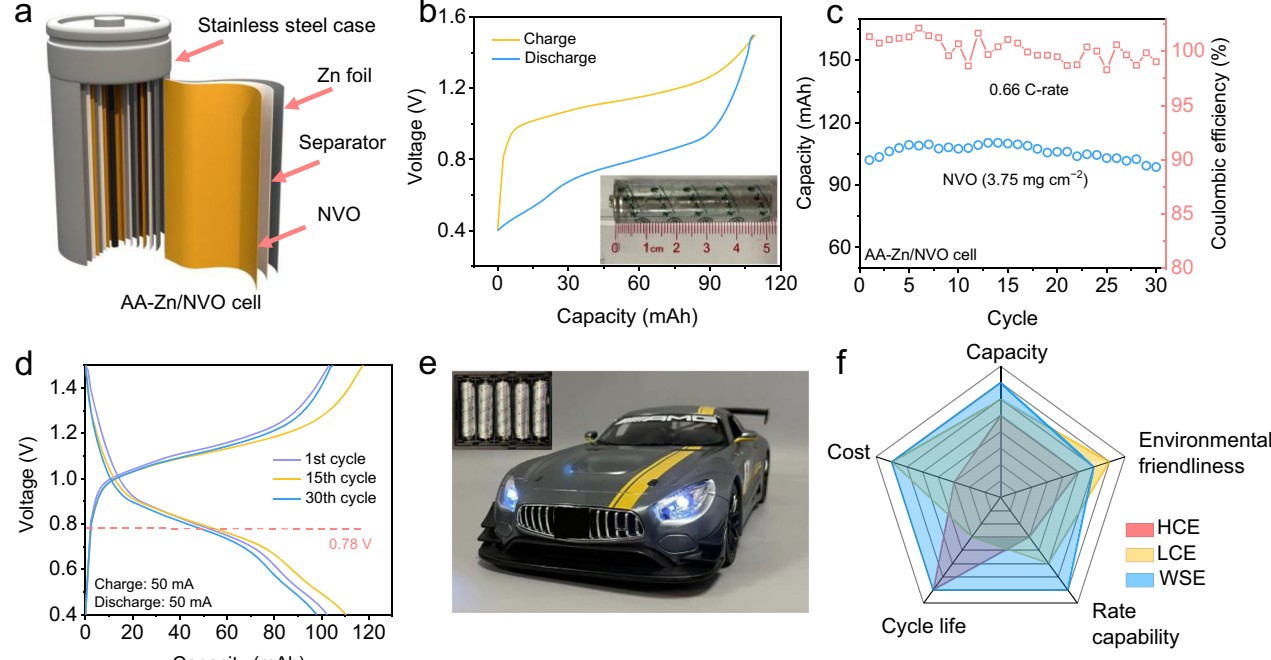

**Fig. 4 | Electrochemical performance of AA-Zn/NVO cells. a** Schematic illustration of AA-Zn/NVO cells. **b** GCD curves at 0.66 C, **c** cycling stability at 0.66 C of AA-Zn/NVO cells and **d** selected GCD curves of 1st, 15th and 30th cycles at 0.66 C. **e** Digital images of remote-controlled car powered by AA-Zn/NVO cells. The insets in Fig. 4b and e are the optical images of as-fabricated AA-Zn/NVO cells. **f** Comparison of WSE-based AZIBs with other reported AZIBs based on other different electrolyte chemistry.

cylinder (Supplementary Fig. 39). Different from the most reported studies relying on laboratory-scale coin cells with high N/P and E/C ratios, much lower N/P and E/C ratios is achieved in this AA-type cell (Supplementary table S4). To avoid potential electrolyte decomposition, the voltage window of AA-Zn/NVO is compressed down to 0.4–1.5 V. A satisfying capacity of 101.7 mAh (339 mAg$^{-1}$) was obtained at 0.66 C-rate, which is closed to the theoretical capacity of NVO cathode and even comparable to some commercial zinc manganese dry battery (Fig. 4b). More importantly, the selected GCD curves of AA-Zn/NVO at the 1st, 15th, and 30th cycles show no obvious capacity loss or voltage hysteresis, proving it can support very stable long cycling as well as high CE (Fig. 4c and d). The NVO and Zn electrodes of the device after cycling assessment were taken out for further SEM characterization. The SEM images of the NVO electrode after cycling show no significant change while Zn foil presents a smooth surface (Supplementary Fig. 40). All the above advantages support our proposed WSE can stabilize both NVO cathode and Zn anode, even in a practical rechargeable AA-type cell system. As a proof of concept, the integrated AA-Zn/NVO with WSE can provide sufficient power to drive a remote-controlled car (Fig. 4e and Supplementary Movie 1). Taking all factors including capacity, rate capability, cycle life and cost into consideration, it is reasonable to expect that such WSE could outperform the most advanced AZIBs designed via other different electrolyte chemistry (Fig. 4f).

## Discussion

In summary, we developed a promising WSE for highly sustainable AZIBs by simultaneously stabilizing the cathode and Zn anode. This was achieved by using butanone as an anti-solvent to lower the solvating power of $H_2O$, and thus reduce the solvated $H_2O$ to form a water-poor solvation structure ($[(Zn^{2+})(H_2O)_{4.3}(SO_4^{2-})_{1.3}(C_4H_8O)_{0.4}]$) to relieve the structure aberration/collapse of cathode during cycling as well as dendrite growth and side reactions on Zn anode. As a proof of concept, excellent stability of the Zn anode at the test condition (60 mA cm$^{-2}$ and 60 mAh cm$^{-2}$, 88.2% DOD) is achieved. Moreover,

three typical AZIBs including Zn/NVO, Zn/MnO$_2$, and Zn/CoFe(CN)$_6$ AZIBs using WSE were assembled and exhibited excellent cycling stability. The effectiveness of WSE for AZIBs is further demonstrated in AA-Zn/NVO, of which a high capacity of 101.7 mAh and high stability of 96.1% capacity retention after 30 cycles were achieved. Although there is still much to explore, such as the search for better-performing cathode and the manufacturing technique for higher mass loading cathode (> 20 mg cm$^{-2}$), this electrolyte design strategy takes the first step toward realizing the high sustainability of AZIBs.

## Methods
### Electrolyte preparation
LCE was prepared by dissolving 5.75 g of ZnSO$_4$·7H$_2$O into 40 mL of deionized water to form transparent solutions. To prepare WSE, 5.75 g of ZnSO$_4$·7H$_2$O is added to a mixed solution containing 5 mL of butanone and 35 mL of deionized water until the solutions become transparent. The concentration of ZnSO$_4$ in WSE is 0.5 M.

### Cathode preparation
NaV$_3$O$_8$·1.5H$_2$O was prepared by adding 2 g of commercial V$_2$O$_5$ into 2 M NaCl aqueous solution (30 mL) under stirring for 4 days at 30 °C. The dark red product was washed with deionized (DI) water several times and obtained after freeze-drying for 48 h. CoFe(CN)$_6$ was prepared by dissolving 0.1 mmol K$_3$[Fe(CN)$_6$] and 0.7 g sodium dodecyl sulfate (SDS) in 20 mL deionized water to form solution A. Then, 20 mL 1 mM Co(CH$_3$COO)$_2$·4H$_2$O aqueous solution is slowly added into solution A. After aging at room temperature for 24 hours, the products are washed several times with DI water. The samples are finally achieved after freeze-drying for 48 h. MnO$_2$ (MA-EN-CA-1U015Y) was purchased from Canrd Technology Co. Ltd. and used without any treatment. Take the NVO electrode as an example, NVO was mixed with carbon black and poly(vinylidene fluoride) binder at a weight ratio of 8:1:1 in 1-methyl-2-pyrrolidinone (NMP). The slurry was pasted on Ti foil (0.3 mm thickness), and dried overnight at 60 °C. The total mass loading of the electrode was 2 mg cm$^{-2}$.

## Fabrication of AA-type battery

The NVO cathode was prepared as the same routine which we have described in the above section. The total mass loading of the electrode was 3.75 mg cm$^{-2}$. Zn foil with 0.3 mm thickness was selected as the anode and brown paper was used as separator. The cathode, separator, and anode were rolled too tight and filled into a stainless steel cylinder. The tabs of NVO cathode and Zn anode were bonded by the tab film (MTI) via hot-pressing. The stainless steel cylinder is sealed after the injection of 2.0 g WSE. The sealing procedure was conducted by the automated vacuum sealing machine (Wuhan Geruisi New Energy Co., Ltd). The obtained AA-type battery was rested for an hour before the electrochemical test.

## Materials characterization

Raman spectroscopy (Renishaw inVia), Fourier-transform infrared spectroscopy (Vertex70Hyperion3000), and nuclear magnetic resonance (Varian INOVA500NB) were used to investigate the solvation structures of Zn$^{2+}$. Differential scanning calorimetry and ionic conductivity of different electrolytes were tested by DSC-204 F1 and Leici DDS-11A. Field-emission SEM (JSM-6330F), XRD (D-MAX 2200 VPC, RIGAKU) TG (TG209F1 Libra R), and XPS (NEXSA, Thermo FS) were performed to study the microstructures and compositions of Zn anode and NVO cathode.

## Electrochemical measurements

CV, GCD, GITT, and EIS measurements were conducted on CHI 760E electrochemical workstation and Neware battery system (CT-3008-5V 10mA-164, Shenzhen, China). For galvanostatic cycling of Zn/Zn symmetric cells, 2032-type coin cells were assembled with Zn foil (N-buliv Co., Ltd., 1 mm thickness) as cathode and anode, glass fiber (Whatman GF/D) as separator. For the Zn plating/stripping test, Zn foil was used as counter electrode, and Ti plate (1 mm thickness) was employed as working electrode. The stripping cut-off voltage was set at 0.5 V (vs. Zn$^{2+}$/Zn) for each cycle. The electrochemical characterizations of the Zn/NVO, Zn/MnO$_2$ and Zn/CoFe(CN)$_6$ cells were performed in 2032-type coin cells or AA-type cells with different electrolytes. The voltage window for Zn/NVO, Zn/MnO$_2$ and Zn/CoFe(CN)$_6$ coin cells are 0.2–1.6 V, 0.8–1.8 V, and 0.8–1.9 V, respectively. For EIS measurement, the applied frequency is between 10$^{-2}$ and 10$^5$ Hz. In the GITT study, a cell was discharged at 500 mA g$^{-1}$ rate for 30 s, followed by a 10 min open circuit step to allow relaxation back to equilibrium. The procedure was continued until the discharge voltage reached 0.2 V. All electrochemical tests were carried out in an environmental chamber at a constant temperature of 25 °C. The energy density (E) and power density (P) of Zn/NVO, Zn/MnO$_2$ and Zn/CoFe(CN)$_6$ coin cells were calculated from GCD curves according to Eqs. (2) and (3), respectively, where I is the discharging current, U is the voltage, and t is the discharging time.

$$E = \frac{I \int U dt}{m} \tag{2}$$

$$P = \frac{E}{t} \tag{3}$$

## Theoretical calculations

Force-field molecular dynamics (FFMD) simulations were performed using the MD software Material Studio. The forced field parameters were obtained from Universal fields. The size of box is 4.4 × 4.4 × 4.4 nm$^3$, and periodic boundary conditions were set in both directions. The simulation cells contained 2200 H$_2$O and 20 ZnSO$_4$ (LCE) and 2200 H$_2$O, 20 ZnSO$_4$ and 4 butanone (WSE), respectively. Ewald methods were applied to compute electrostatic interactions. For the calculation of electrostatic/non-electrostatic interactions in real space, the cut-off length was set as 0.6 nm and the integration time

step was 1 fs. The system was annealed from 200 to 500 K over 0.5 ns and then run for 2.0 ns to reach equilibrium. Temperature simulations were performed using the Nose method. For the post-processing analysis, a 5 ns production simulation was performed.

## Data availability

All data that support the findings of this study are provided within the paper and its Supplementary Information. All additional information is available from the corresponding authors upon request. Source data are provided with this paper.

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

## Acknowledgements

This work is financially supported by the Guangdong Basic and Applied Basic Research Foundation (2022A1515140008), and the Guangdong Province Innovation and Strong School Project (2020ZDZX2004).

## Author contributions

X.L. planned and designed the project. Z.Y., X.S, and J.W. fabricated the materials and performed the electrochemical experiments. J.X. analyzed the data. X.L., X.S., Z.Y., and S.X. wrote the manuscript. All authors discussed the results and commented on the manuscript.

## Competing interests

The authors declare no competing interests.
