## [Peer Review File · Nature Communications]

A weakly solvating electrolyte towards practical rechargeable aqueous zinc-ion batteriesREVIEWER COMMENTS

Reviewer #1 (Remarks to the Author):

In this manuscript, Shi et al. reported a weakly solvating electrolyte (WSE) to solve the current critical issues of AZIBs. The proposed WSE enables three representative full cells with improved cycling stability compared to low concentrated electrolyte (LCE). The protection mechanism of WSE is also discussed and supported by solid data. Moreover, an AA-type Zn//NVO cell is successfully fabricated, which is very impressive. Therefore, I suggest this work can be published in Nat. Commun. after a suitable revision.

1. Is 12.5% v/v butanone the optimized ratio of WSE? What about the different concentrations of Zn salts?
2. More specific measurements should be used to investigate the electrolyte, e.g. DSC and viscosity.
3. EIS spectra of Zn//NVO full cells based on different electrolytes and related discussion should be provided.
4. It is well known V-based materials face dissolution problem in the aqueous electrolyte. The authors should also investigate whether WSE can suppress the dissolution of V-based materials and provide related evidence and discussion.
5. The authors only provide Zn//NVO cells at high mass loading situation. What is the cycling stability of other two cells (Zn//CoFe(CN)₆ and Zn//MnO₂ cells) with high cathodic mass loading?
6. It is suggested to compare the electrochemical performance of Zn//NVO cell in this work with other recently reported works to highlight the effectiveness of WSE.
7. The Ragone plots of Zn//NVO, Zn//CoFe(CN)₆ and Zn//MnO₂ cells based on WSE should be provided.
8. Butanone shows much lower freezing point (−85.9 °C) than water. So can a Zn//NVO full cell using WSE work at low temperature condition, e.g. −20 °C?
9. The detailed experimental information should be provided, such as the thickness of Ti electrode for Zn plating/stripping test, mass loading of NVO electrode for AA-type cell.

Reviewer #2 (Remarks to the Author):

The work by Shi et al. reports an electrolyte organic additive for aqueous Zn-ion batteries. This is another study among many in recent years aimed to mitigate the corrosion of Zn anode. The section of butanone additive as a solvation breaker is well reasoned and the experimental results appear to support the concept. However, the discussion on the role of butanone in reducing desolvation energy barrier at both electrode/electrolyte interfaces is weak. Below is a list of the concerns conceived by this reviewer.

Major concerns:

- It is unclear why low concentration (0.5M) ZnSO₄ was chosen as the base electrolyte. The ionic conductivity of 0.5M ZnSO₄ is only half the commonly used 2M ZnSO₄ electrolyte.
- From the classical theory of solution chemistry, highly solvated electrolytes such as ZnSO₄ have much higher solubility and ionic conductivity than those weakly solvated electrolytes such as ZnAc₂. In this study, since butanone is used to reduce the solvation of the electrolyte, the resultant electrolyte should exhibit lower solubility and conductivity. Instead, the authors showed a higher conductivity. Although the MD simulations suggested the butanone-added ZnSO₄ has a higher MSD, the physical origin of higher conductivity is not given.
- The discussion of the effect of butanone on cathode performance is unsound. It is well

known that molecular water insertion into the layers of the cathode is beneficial to the $\text{Zn}^{2+}/\text{H}^{+}$ insertion as the former enlarges the gallery space for easy $\text{Zn}^{2+}/\text{H}^{+}$ insertion. However, the authors' explanation is the opposite, i.e., butanone pulls H_2O away from the cathode. The authors did not provide solid evidence to support the assertion (only showed TGA with small difference between the two samples, which could be caused by the sample preparation). The authors should perform XRD on samples soaked for different times to see if indeed butanone-added ZnSO_4 has smaller gallery space.

- Following the above comment, the better full cell performance exhibited with butanone-added electrolyte could mainly derive from lower anode overpotential, not necessarily from lowered desolvation energy barrier at the cathode. There is no evidence presented to support that butanone is helping $\text{Zn}^{2+}/\text{H}^{+}$ insertion.

Minor concerns:

- The Zn dendrite growth should be a minor issue for mildly acidic ZnSO_4 electrolyte.
- The bulk properties such as pH, solubility, of the two electrolytes, should also be provided.
- Explain why the Raman shift induced by butanone is so subtle (977.1 vs. 977.3 cm^{-1}) in Fig. 1b, while its impact on the solvation of the electrolyte is so pronounced.
- What is the Raman shift at lower wavenumbers for WSE? See Fig. S2.
- It is unproven to call the slope of Fig. 2f desolvation energy as it is in fact the plotted charge transfer resistance of Zn/electrolyte interface measured from Zn/electrolyte/Zn symmetrical cell. These are two different terms.
- The preparation procedure for butanone added electrolyte is vague. The volume of 0.5M ZnSO_4 to which 5 ml and 35 ml water was added should be specified. Otherwise, it is difficult for others to reproduce the experiment.
- Where is MEK (line 366) coming from in the theoretical calculations section?

Reviewer #3 (Remarks to the Author):

The authors introduced butanone as an additives for electrolyte in zinc ion batteries to effectively improve structure deterioration and undesirable reactions caused by solvated H_2O and greatly increased the cycle performance. We don't have any doubts about influences of butanone on cycle stability, kinetics of zn^{2+} ions, and how the authors analyzed the characteristics of butanone additives. Furthermore, it is interesting that the electrochemical tests of the WSE electrolyte adopted zinc-ion battery exhibited significantly high current density of 60 mA cm^{-2} , outperforming the previously reported zinc-ion batteries, and demonstrated its feasibility for practical applications.

Thus, it is acceptable for publication in this journal after minor revision. Detailed comments are as follows:

1. The information about the abbreviation "HBs" on line 117 cannot be found
2. Page 5, "The solvation structure variation of Zn^{2+} in WSE are also testified by he butanone-induced negative shift of $\nu(\text{SO}_4^{2-})$ band..."
When the Figure 1c is compared to above statement, the expression "negative shift" seems incorrect.
3. Page 10, "The positive shift of Zn 2p spectra..."
Due to slight shift of peaks in Supplementary Figure 23, it is difficult to identify the difference. Authors should consider using vertical lines.
4. Authors should give more detailed explanation of Fig. 3g. 'Pristine material' and 'baseline electrolyte' are used for the first time in Fig. 3g and may confuse the readers. The statement that "The (001) and (110) planes deviate less from..." should be more clearly described so

that the comparison target can be identified.

5. Among the studies that reported weakly solvating electrolytes in the zinc-ion battery, what is the novelty of this study?

6. How the author controlled and decided the proper addition amount of butanone additives?

We thank the reviewer for his/her careful review of our manuscript, and really appreciate the constructive comments. Note that all the changes/additions are red-highlighted in the revised version of the manuscript. Please see below for our detailed responses to the comments.

To reviewer #1:

Comments: In this manuscript, Shi et al. reported a weakly solvating electrolyte (WSE) to solve the current critical issues of AZIBs. The proposed WSE enables three representative full cells with improved cycling stability compared to low concentrated electrolyte (LCE). The protection mechanism of WSE is also discussed and supported by solid data. Moreover, an AA-type Zn//NVO cell is successfully fabricated, which is very impressive. Therefore, I suggest this work can be published in Nat. Commun. after a suitable revision.

Response: Thank you very much for your positive comments. We have revised our manuscript according to your suggestions. Details please see the followings or our revised manuscript.

1. Is 12.5% v/v butanone the optimized ratio of WSE? What about the different concentrations of Zn salts?

Response: Thanks very much for your question. The mixed solution of 0.5 M ZnSO₄ and 12.5% v/v butanone is the optimized WSE, as proved by the comparison of cycling performance of Zn//NVO batteries using butanone-saturated electrolytes with various concentrated ZnSO₄ from 0.3 M to 2 M. As shown in Fig. R1, Zn//NVO batteries with 0.5 M ZnSO₄+12.5% v/v butanone WSE shows the best cycling stability (~97.8% capacity retention after 1000 cycles), outperforming Zn//NVO batteries with other concentrations of Zn salts.

Fig. R1 Cycling stability of Zn//NVO batteries using butanone-saturated electrolytes with varied concentrated ZnSO₄ from 0.3 M to 2 M at 10 A g⁻¹.

2. More specific measurements should be used to investigate the electrolyte, e.g. DSC and viscosity.

Response: We performed DSC and viscosity test to further investigate the WSE. As shown in Fig. R2, the freezing point of WSE is decreased to $-25.8\text{ }^{\circ}\text{C}$, indicating the enhanced anti-freezing property of WSE. The viscosity of LCE and WSE are 1.25 and 1.68 mP s^{-1} , respectively. We have added the DSC and viscosity results of LCE and WSE to the revised manuscript.

Fig. R2 DSC curves of LCE and WSE under a cooling rate of $5\text{ }^{\circ}\text{C min}^{-1}$.

3. EIS spectra of Zn//NVO full cells based on different electrolytes and related discussion should be provided.

Response: Thanks very much for your valuable suggestion. The EIS spectra of Zn//NVO batteries with LCE and WSE after activation is shown in Fig. R3. Zn//NVO battery with WSE shows smaller Warburg resistance (R_w) than that in LCE, indicating the facilitated ion transfer process for Zn^{2+} ion storage due to the changed solvation structure of Zn^{2+} . The EIS spectra and corresponding discussion have been added to the revised manuscript.

Fig. R3 EIS spectra of Zn//NVO full battery based on LCE and WSE.

4. It is well known V-based materials face dissolution problem in the aqueous electrolyte. The authors should also investigate whether WSE can suppress the dissolution of V-based materials and provide related evidence and discussion.

Response: Thanks very much for your academic comment. To investigate whether WSE can suppress the dissolution of V-based materials, NVO electrodes were first immersed in LCE and WSE for 7 days to test its solubility. As shown in Fig. R4a, the bottle containing WSE shows lighter yellow compared with that of LCE, indicating the dissolution of NVO is minished by butanone. To highlight the negative impact of cathode dissolution on the overall performance of Zn//ZVO, NVO electrodes that underwent the dissolution tests (resting the Zn//NVO batteries with LCE and WSE for 7 days) were subjected to GCD tests. Compared to the batteries that did not experience the dissolution test (Fig. 3a), the NVO electrodes tested in LCE and WSE show capacity loss of 44.4 % and 7.8% in the first cycle, respectively, indicating WSE can suppress the cathode dissolution of NVO (Fig. R4b).

Fig. R4 **a** Optical images of NVO electrode immersed in LCE and WSE for 7 days. **b** The first discharge curves of Zn//NVO batteries after resting for 144 h in LCE and WSE.

5. The authors only provide Zn//NVO cells at high mass loading situation. What is the cycling stability of other two cells (Zn//CoFe(CN)₆ and Zn//MnO₂ cells) with high cathodic mass loading?

Response: Thanks for your comment. The cycling stability of Zn//CoFe(CN)₆ and Zn//MnO₂ batteries with high cathodic mass loading is presented in Fig. R5. Compared to batteries using LCE, extended capacity retention of 74.4% and 57.9% can be obtained for Zn//CoFe(CN)₆ and Zn//MnO₂ batteries using WSE.

Fig R5. Cycling stability of high mass loading **a** Zn//CoFe(CN)₆ and **b** Zn//MnO₂ batteries using LCE and WSE at 0.1 A g⁻¹.

6. It is suggested to compare the electrochemical performance of Zn//NVO cell in this work with other recently reported works to highlight the effectiveness of WSE.

Response: To highlight the effectiveness of WSE, we compare the cycling stability of Zn//NVO batteries with other recently reported works focusing on electrolyte design. As shown in Table R1, the capacity retention of Zn//NVO batteries using WSE is higher than that of Zn//V batteries based on other electrolyte chemistry, confirming the effectiveness of WSE.

Table R1. Comparison of the electrochemical stability of Zn-V battery with different electrolyte chemistry.

Cathode	Anode	Electrolyte	Capacity retention	Ref.
NVO	Zn foil	0.5 M ZnSO ₄ + 12.5 v/v% butanone	97.6% after 1000 cycles at 5 A g ⁻¹	This work
V ₂ O ₅ /graphene oxide	Zn foil	21 m LiTFSI + 3 m ZnOTf ₂ +10 wt % PVA	93% over 600 cycles at 500 mA g ⁻¹	ACS Appl. Mater. Interfaces, 2020, 12, 15305
V ₂ O ₅	Zn foil	2 M ZnSO ₄ + 25% sulfolane	70% after 500 cycles at 10 A	Angew. Chem. Int. Ed., 2023,

			g^{-1}	62, e202214966
V_2O_5	Zn foil	$\text{Zn}(\text{OTf})_2 + 30\%$ 2-propanol	79.6% after 1500 cycles at 2 A g^{-1}	Adv. Mater., 2022, 34, 2207344
$\text{V}_2\text{O}_5 \cdot n\text{H}_2\text{O}$	Zn foil	2 M $\text{Zn}(\text{OTf})_2 + 40\%$ DME	93.1% after 2000 cycles at 2 A g^{-1}	Energy Storage Mater., 2022, 47, 203
NVO	Zn foil	2 M $\text{ZnSO}_4 + 0.1$ M ImS	88% after 3000 cycles at 20 A g^{-1}	Energy Environ. Sci., 2022, 15, 4748
$\text{Zn}_x\text{V}_2\text{O}_5 \cdot n\text{H}_2\text{O}$	Zn foil	3 M $\text{ZnSO}_4 + 68\%$ EG	89.6% after 500 cycles at 500 mA g^{-1}	Nano Energy, 2021, 80, 105478

7. The Ragone plots of Zn//NVO, Zn//CoFe(CN)₆ and Zn//MnO₂ cells based on WSE should be provided.

Response: The Ragone plots of Zn//NVO, Zn//CoFe(CN)₆ and Zn//MnO₂ batteries using WSE are shown in Fig. R6 (based on the mass loading of cathode materials), in which energy densities of 269, 170.5 and 360.2 Wh kg^{-1} are obtained for Zn//NVO, Zn//CoFe(CN)₆ and Zn//MnO₂ batteries.

Fig. R6 Ragone plots of Zn//NVO, Zn//CoFe(CN)₆ and Zn//MnO₂ batteries using WSE.

8. Butanone shows much lower freezing point ($-85.9\text{ }^\circ\text{C}$) than water. So can a Zn//NVO full cell using WSE work at low temperature condition, e.g. $-20\text{ }^\circ\text{C}$?

Response: Thanks for your comment. The Zn//NVO batteries can maintain a capacity of 178.7 mAh g⁻¹ after 100 cycles with a capacity retention of 66.4% at -20 °C (Fig. R7), indicating the wide working temperature range of WSE.

Fig. R7 Low temperature cycling performance of Zn//NVO batteries at 0.1 A g⁻¹.

9. The detailed experimental information should be provided, such as the thickness of Ti electrode for Zn plating/stripping test, mass loading of NVO electrode for AA-type cell.

Response: Thanks for your comment. We have replenished the experimental information.

To reviewer #2:

Comments: The work by Shi et al. reports an electrolyte organic additive for aqueous Zn-ion batteries. This is another study among many in recent years aimed to mitigate the corrosion of Zn anode. The section of butanone additive as a solvation breaker is well reasoned and the experimental results appear to support the concept. However, the discussion on the role of butanone in reducing desolvation energy barrier at both electrode/electrolyte interfaces is weak. Below is a list of the concerns conceived by this reviewer.

Response: Thank you very much for your comments. We have revised our manuscript according to your suggestions. Details please see the followings or our revised manuscript.

Major concerns:

- *It is unclear why low concentration (0.5M) ZnSO₄ was chosen as the base electrolyte. The ionic conductivity of 0.5M ZnSO₄ is only half the commonly used 2M ZnSO₄ electrolyte.*

Response: Thank you for your valuable comments. The tested ionic conductivity of 0.5M ZnSO₄ and 2M ZnSO₄ are 26.3 and 46.7 S cm⁻¹, respectively. Considering ionic conductivity will affect the electrochemical performance of battery, we have investigated the influence of ZnSO₄ concentration on the electrochemical properties of Zn//NVO battery. As shown in Fig. R8a and b, the Zn//NVO full battery with 0.5 M and 2 M ZnSO₄ shows similar capacity at 0.1 A g⁻¹ (350.1 mAh g⁻¹ and 321.5 mAh g⁻¹) and rate performance at 10 A g⁻¹ (49.4% and 43.9% capacity retention). Moreover, the mixed solution of 0.5 M ZnSO₄ and 12.5% v/v butanone exhibits the best cycling performance (~97.8% capacity retention after 1000 cycles), indicating the optimized WSE formula is 0.5 M ZnSO₄ + 12.5% v/v butanone (Fig. R8c and d). Based on the slight electrochemical performance difference between Zn//NVO full battery with 0.5 M and 2 M ZnSO₄, and the significantly improved cycling stability of Zn//NVO battery using 0.5 M ZnSO₄ and 12.5% v/v butanone, the 0.5 M ZnSO₄ was chosen as the base electrolyte for the intuitive comparison. The optimization of electrolyte and corresponding discussion have been added to the revised manuscript.

Fig. R8 Rate capability of Zn/NVO batteries using **a** 2 M ZnSO₄ and **b** 0.5 M ZnSO₄. Cycling performance of Zn/NVO batteries using **c** LCE and **d** WSE with different concentration.

• From the classical theory of solution chemistry, highly solvated electrolytes such as ZnSO₄ have much higher solubility and ionic conductivity than those weakly solvated electrolytes such as ZnAc₂. In this study, since butanone is used to reduce the solvation of the electrolyte, the resultant electrolyte should exhibit lower solubility and conductivity. Instead, the authors showed a higher conductivity. Although the MD simulations suggested the butanone-added ZnSO₄ has a higher MSD, the physical origin of higher conductivity is not given.

Response: Thank you very much for your kind reminder. We have re-tested the ionic conductivity of the LCE and WSE and found that the previous data is incorrect, which may be due to the uncalibrated conductivity meter. The re-tested ionic conductivity for WSE is about 18.1 S cm⁻¹, lower than that of LCE (26.3 S cm⁻¹, Fig. R9). As the ionic conductivity consists of the mobility of both anion and cations, the reason for the lower ionic conductivity of WSE may be the reduced anion mobility due to the introduction of butanone that shows hydrogen bonding effect with SO₄²⁻, while the Zn²⁺ transference number and Zn²⁺ diffusion coefficient is increased in WSE compared to that in LCE. We have corrected the ionic conductivity of WSE and added the related discussion in the revised manuscript.

Fig. R9 Ionic conductivity of LCE and WSE.

• *The discussion of the effect of butanone on cathode performance is unsound. It is well known that molecular water insertion into the layers of the cathode is beneficial to the Zn^{2+}/H^+ insertion as the former enlarges the gallery space for easy Zn^{2+}/H^+ insertion. However, the authors' explanation is the opposite, i.e., butanone pulls H_2O away from the cathode. The authors did not provide solid evidence to support the assertion (only showed TGA with small difference between the two samples, which could be caused by the sample preparation). The authors should perform XRD on samples soaked for different times to see if indeed butanone-added $ZnSO_4$ has smaller gallery space.*

Response: Thanks for your comments. We agree that the insertion of H_2O molecules with a suitable amount into the layers of cathode is beneficial to the Zn^{2+}/H^+ insertion. However, previous works revealed that V_2O_5 experiences a dramatic lattice variation as overdosed water molecules intercalate into its layered spacing, which is a potential reason for the capacity fading (*ACS Energy Lett.* 2018, 3, 1366; *Adv. Sci.* 2021, 8, 2102053). In our case, we also found that H_2O molecules can insert into both NVO electrodes cycled in LCE and WSE, as suggested by the weight loss above 200 °C in TG measurement. The NVO electrode cycled in WSE shows less loss of lattice H_2O , indicating the co-inserted H_2O is decreased to some extent due to the reduced solvated H_2O in the primary solvation sheath of Zn^{2+} . In addition, XRD patterns of NVO electrode after 200 cycles in LCE experiences a more obvious lattice expansion than that in WSE, as revealed by the less deviated (001) plane from the pristine NVO in Fig. 3g. Therefore, it is concluded the co-insertion of H_2O is partially suppressed in WSE due to the formation of CIP solvation structure of Zn^{2+} , leading to the relieved structure aberration/collapse during cycling in this case.

Moreover, we performed XRD on samples soaked in LCE and WSE for 7 days to investigate the function of butanone in WSE. As shown in Fig. R10, the (001) and (110) planes of NVO soaked

in both LCE and WSE shows no deviate from the pristine NVO except the (001) plane of NVO soaked in LCE is missing, indicating the structure of NVO is destroyed in LCE. Combined with the aforementioned dissolution tests in Fig. R4, such structure degradation of NVO soaked in LCE is mainly due to the dissolution issue. We have added the related discussion in the revised manuscript.

Fig. R10 XRD patterns of NVO after soaking in LCE and WSE for 7 days.

- *Following the above comment, the better full cell performance exhibited with butanone-added electrolyte could mainly derive from lower anode overpotential, not necessarily from lowered desolvation energy barrier at the cathode. There is no evidence presented to support that butanone is helping Zn^{2+}/H^+ insertion.*

Response: Thank you very much for your comments. The positive effect of WSE for helping Zn^{2+} insertion is proved by the GITT test, of which the ion diffusion coefficient in WSE at second plateau representing Zn^{2+} insertion is about one order of magnitude higher than that in LCE, indicating the boosted kinetics of Zn^{2+} insertion (Fig. R11 or Fig. 3i). In addition, the capacitance contribution at all scan rates of Zn//NVO battery using WSE are higher than that in pure $ZnSO_4$ electrolytes, suggesting the faster reaction kinetics of NVO in WSE (Supplementary Fig. 34).

Fig. R11 Diffusivity coefficient at various states of discharge in LCE and WSE according to GITT measurement.

Minor concerns:

- *The Zn dendrite growth should be a minor issue for mildly acidic ZnSO₄ electrolyte.*

Response: Thanks for your comment. We agree with this reviewer that Zn dendrite growth is not severe in mildly acidic ZnSO₄ electrolyte, only when the depth of discharge (DOD) of tested Zn anode is low. Take this work for example, the dendrite growth of Zn anode at 1 mAh cm⁻² (DOD=1.7%) is mild, as proved by the cycling stability of Zn//Zn symmetrical battery with LCE (~70 h). However, when the test condition increased to 60 mAh cm⁻² (DOD=88.2%), the Zn//Zn symmetrical battery with LCE cannot work for a cycle, indicating its severe dendrite growth problem. In contrast, Zn//Zn symmetrical battery using WSE can work for more than 70 h at 60 mAh cm⁻², indicating the dendrite growth can be relieved in WSE.

- *The bulk properties such as pH, solubility, of the two electrolytes, should also be provided.*

Response: The pH and solubility of the two electrolytes have been provided according to your suggestions. The pH of LCE and WSE is 4.28 and 4.02. The solubility of butanone in 0.5 M ZnSO₄ is 12.5 v/v%. We have added the above information in the revised manuscript.

- *Explain why the Raman shift induced by butanone is so subtle (977.1 vs. 977.3 cm⁻¹) in Fig.1b, while its impact on the solvation of the electrolyte is so pronounced.*

Response: Thanks for your comments. Indeed, the Raman shift induced by butanone is subtle. However, when overlapping the Raman spectra of LCE and WSE, an obvious difference at a range of 985.8-1008.9 cm⁻¹ can be observed, which represents the CIP solvation structure of Zn²⁺ (Fig. R12, *Angew. Chem. Int. Ed.* 2020, 59, 9377–9381). We have added the more detailed discussion of Fig. 1b in the revised manuscript.

Fig. R12 Overlapped Raman spectra of LCE and WSE.

- *What is the Raman shift at lower wavenumbers for WSE? See Fig. S2.*

Response: Thanks for your question. The peaks at 2888 and 2941 cm^{-1} are corresponding to $-\text{CH}_3$ stretching vibration of butanone while the peaks at 2925 cm^{-1} is corresponding to $-\text{CH}_2-$ stretching vibration of butanone. We have added the related information in the revised supplementary information.

- *It is unproven to call the slope of Fig. 2f desolvation energy as it is in fact the plotted charge transfer resistance of Zn/electrolyte interface measured from Zn/electrolyte/Zn symmetrical cell. These are two different terms.*

Response: Thank you very much for your reminder. We have corrected the appellation of Fig. 2f as the plotted charge transfer resistance of Zn/electrolyte interface measured from Zn//Zn symmetrical battery.

- *The preparation procedure for butanone added electrolyte is vague. The volume of 0.5M ZnSO_4 to which 5 ml and 35 ml water was added should be specified. Otherwise, it is difficult for others to reproduce the experiment.*

Response: Thank you very much for your question. To prepare WSE, 5.75 g of $\text{ZnSO}_4 \cdot 7\text{H}_2\text{O}$ is added to a mixed solution containing 5 mL of butanone and 35 mL of deionized water until the solutions becoming transparent. The concentration of ZnSO_4 in the above solution is 0.5 M. We have added the above information to the revised manuscript.

- *Where is MEK (line 366) coming from in the theoretical calculations section?*

Response: Thanks for your question. MEK appeared there by mistake. We have corrected it as butanone. We further checked the whole manuscript carefully.

To reviewer #3:

Comments: The authors introduced butanone as an additive for electrolyte in zinc ion batteries to effectively improve structure deterioration and undesirable reactions caused by solvated H₂O and greatly increased the cycle performance. We don't have any doubts about influences of butanone on cycle stability, kinetics of Zn²⁺ ions, and how the authors analyzed the characteristics of butanone additives. Furthermore, it is interesting that the electrochemical tests of the WSE electrolyte adopted zinc-ion battery exhibited significantly high current density of 60 mA cm⁻², outperforming the previously reported zinc-ion batteries, and demonstrated its feasibility for practical applications. Thus, it is acceptable for publication in this journal after minor revision.

Detailed comments are as follows:

Response: Thank you very much for your positive comments. We have revised our manuscript according to your suggestions. Details please see the followings or our revised manuscript.

1. The information about the abbreviation "HBs" on line 117 cannot be found.

Response: Thanks for your reminder. The abbreviation "HBs" represents "hydrogen bonds". We added the information in the revised manuscript.

2. Page 5, "The solvation structure variation of Zn²⁺ in WSE are also testified by the butanone-induced negative shift of $\nu(\text{SO}_4^{2-})$ band..." When the Figure 1c is compared to above statement, the expression "negative shift" seems incorrect.

Response: Thank you very much for your reminder. We are sorry for the incorrection of the description of Fig. 1c. It should be "The solvation structure variation of Zn²⁺ in WSE are also testified by the butanone-induced positive shift of $\nu(\text{SO}_4^{2-})$ band in Fourier transform infrared spectroscopy (FT-IR) (Fig. 1c)". We corrected it in the revised manuscript.

3. Page 10, "The positive shift of Zn 2p spectra..."Due to slight shift of peaks in Supplementary Figure 23, it is difficult to identify the difference. Authors should consider using vertical lines.

Response: Thanks for your comment. We redrew Supplementary Figure S23 and used vertical lines to highlight the peak shift (Fig. R13).

Fig. R13 Zn 2p XPS spectra of NVO electrode at fully discharged and charged state in WSE electrolyte.

4. Authors should give more detailed explanation of Fig. 3g. ‘Pristine material’ and ‘baseline electrolyte’ are used for the first time in Fig. 3g and may confuse the readers. The statement that “The (001) and (110) planes deviate less from...” should be more clearly described so that the comparison target can be identified.

Response: Thank you very much for your suggestion. The ‘Pristine material’ and ‘baseline electrolyte’ in Fig. 3g. represents pristine NVO electrode material and LCE. The statement that “The (001) and (110) planes deviate less from...” was rewritten as “The (001) planes of NVO cycled in LCE and WSE are located at 12.1° and 12.0°. Compared to NVO cycled in LCE, the (001) plane of NVO cycled in WSE show less deviated from the pristine NVO (10.8°), suggesting the mitigatory lattice variation of NVO in WSE.” in the revised manuscript.

5. Among the studies that reported weakly solvating electrolytes in the zinc-ion battery, what is the novelty of this study?

Response: Thank you very much for your question. To the best of our knowledge, there is only few reported works about weakly solvating electrolytes for aqueous Zn ion battery, which focus on the function of WSE on Zn anode protection. Different from it, this work proposes a new WSE formula to concurrently solve the structure degradation and side reaction on both cathode and anode. Consequently, Zn anode exhibits outstanding cycling performance in WSE at ultrahigh current density and areal capacity (60 mA cm^{-2} and 60 mAh cm^{-2}) with high depth of discharge (88.2%), far better than the state-of-art Zn anodes in previous works. Besides, three representative AZIBs including Zn// $\text{NaV}_3\text{O}_8 \cdot 1.5\text{H}_2\text{O}$, Zn// MnO_2 and Zn// $\text{CoFe}(\text{CN})_6$ batteries all shows significantly enhanced cycling stability, confirming the universality of WSE for the stabilization of AZIBs.

Moreover, the effectiveness of this strategy is further and for the first time demonstrated in practical rechargeable AA type Zn//NaV₃O₈·1.5H₂O batteries, which shows a high capacity of 101.7 mAh and outstanding stability of 96.1% capacity retention after 30 cycles at 0.66 C.

6. How the author controlled and decided the proper addition amount of butanone additives?

Response: Thank you very much for your question. We firstly investigated the solubility of butanone in 0.5 M ZnSO₄ solution. As obvious delamination is observed in 0.5 M ZnSO₄ with 15 v/v% butanone, we prepared three kinds of colorless, transparent electrolyte mixture, 0.5 M ZnSO₄ with 5 v/v% butanone, 0.5 M ZnSO₄ with 10 v/v% butanone, and 0.5 M ZnSO₄ with 12.5 v/v% butanone, for further study (Fig. R14a). The electrochemical performance of Zn//NVO battery with different electrolytes is shown in Fig. R14b, in which the full battery with 0.5 M ZnSO₄ with 12.5 v/v% butanone exhibits the best cycling performance (~97.8% capacity retention after 1000 cycles). Therefore, the addition of 12.5% v/v butanone were chosen as the proper amount.

Fig. R14 a Optical image of 0.5 M ZnSO₄ with different addition amount of butanone. **b** Cycling performance of Zn//NVO batteries using 0.5 M ZnSO₄ with different addition amount of butanone.

REVIEWER COMMENTS

Reviewer #1 (Remarks to the Author):

The authors have addressed all previous comments nicely, the paper can be accepted in the current form.

Reviewer #2 (Remarks to the Author):

This resubmitted paper provided some new data and explanations to the reviewer's early questions. However, key fundamental questions on how butanone protects cathode are still not addressed satisfactorily. Like many publications in the literature have demonstrated, adding electron withdrawing organic molecules into aqueous electrolytes will help protect Zn-anode from corrosion. The authors have also shown it in symmetrical cells (Fig.2). This is not a surprise. However, the benefit of butanone additive to protect cathode and the corresponding explanations are still not convincing. The long-term cycle stability and capacity level with WSE are not competitive enough to many Zn-ion systems published, whether with additive or not. More importantly, the explanations to why and how butanone protects cathode are not convincing and in fact contradictory to the experimental results. For example, the color change in both LCE and WSE to yellowish is a strong indicator of V-oxide dissolution, which is contradictory to the conclusion that butanone stops dissolution. In V-chemistry, the appearance of low 2-theta (001) peak is an indicative of formation of hydrated compound with layered structure in which H₂O is situated; this structure is favorable to Zn²⁺/H⁺ storage. However, Fig.3g of TGA shows the opposite. In addition, the 2-theta values related to (001) plane for NVO cycled in both LCE and WSE is within the error bar, 12.1° vs. 12.0°, implying that there is virtually no impact from butanone on cathode protection. Having said the above, it appears that the degradation shown in Fig. R8c of LCE full cell originates from the anode corrosion and cathode dissolution, and butanone's role in WSE is to protect anode. Since lower concentration of ZnSO₄ has a higher pH, it will corrode the anode more severely. Fig.R8c indeed shows faster degradation and lower capacity at lower ZnSO₄ concentrations. Overall, the paper falls short of showing compelling evidence that WSE can protect cathode, which is a pressing issue for all ongoing electrolyte engineering efforts.

Other comments:

- pH value for 0.5M ZnSO₄ is questionable. It appears much lower than the literature data as well this reviewer's own experience (pH~5.0), which prompts the question if other pH value is accurate, or pH meter is calibrated before each measurement??
- There is no explanation on why adding butanone to ZnSO₄ reduces pH.
- Fig R11 (Fig.3i) only shows higher D at higher DoD, why? How accurate is GITT method at high DoD?
- Fig.3c, cycle stability at 5A/g should be tested for more than thousands of hours data to be competitive with literature data that have shown systems better stability even without additive.

Reviewer #3 (Remarks to the Author):

The author have nicely addressd the question and provided the proper response. I suggest the acceptance for publication in Nature commnun.

We thank the reviewer for his/her careful review of our manuscript, and really appreciate the constructive comments. Note that all the changes/additions are red-highlighted in the revised version of the manuscript. Please see below for our detailed responses to the comments.

To reviewer #2:

This resubmitted paper provided some new data and explanations to the reviewer's early questions. However, key fundamental questions on how butanone protects cathode are still not addressed satisfactorily. Like many publications in the literature have demonstrated, adding electron withdrawing organic molecules into aqueous electrolytes will help protect Zn-anode from corrosion. The authors have also shown it in symmetrical cells (Fig.2). This is not a surprise. However, the benefit of butanone additive to protect cathode and the corresponding explanations are still not convincing. The long-term cycle stability and capacity level with WSE are not competitive enough to many Zn-ion systems published, whether with additive or not. More importantly, the explanations to why and how butanone protects cathode are not convincing and in fact contradictory to the experimental results. For example, the color change in both LCE and WSE to yellowish is a strong indicator of V-oxide dissolution, which is contradictory to the conclusion that butanone stops dissolution. In V-chemistry, the appearance of low 2-theta (001) peak is an indicative of formation of hydrated compound with layered structure in which H₂O is situated; this structure is favorable to Zn²⁺/H⁺ storage. However, Fig.3g of TGA shows the opposite. In addition, the 2-theta values related to (001) plane for NVO cycled in both LCE and WSE is within the error bar, 12.1° vs. 12.0°, implying that there is virtually no impact from butanone on cathode protection. Having said the above, it appears that the degradation shown in Fig. R8c of LCE full cell originates from the anode corrosion and cathode dissolution, and butanone's role in WSE is to protect anode. Since lower concentration of ZnSO₄ has a higher pH, it will corrode the anode more severely. Fig.R8c indeed shows faster degradation and lower capacity at lower ZnSO₄ concentrations. Overall, the paper falls short of showing compelling evidence that WSE can protect cathode, which is a pressing issue for all ongoing electrolyte engineering efforts.

Response: We sincerely acknowledge your valuable comments on our work. We have carried out additional experiments to prove the positive effect of WSE for cathode protection and revised our manuscript accordingly. Details please see the followings or our revised manuscript.

Response to the comment “*The long-term cycle stability and capacity level with WSE are not competitive enough to many Zn-ion systems published, whether with additive or not.*”:

The cycle numbers of long-term cycling test of Zn/NVO cells with LCE and WSE is extended to 20000 cycles, which is among the longest cycle number of reported V-based aqueous Zn ion batteries. As shown in Fig. R1, Zn/NVO cell with WSE exhibits a capacity retention of 99.1% while that with LCE fails quickly after only 400 cycles. Additionally, the cycling stability of Zn/NVO cells with WSE is better than that of Zn/V cells based on other electrolyte chemistry, confirming the effectiveness of WSE. (Table R1).

Fig. R1 Cycling stability comparison at 5 A g⁻¹ of Zn/NVO cells in LCE and WSE.

Table R1. Comparison of the electrochemical stability of Zn-V battery with different electrolyte chemistry.

Cathode	Anode	Electrolyte	Capacity retention	Ref.
NVO	Zn foil	0.5 M ZnSO ₄ + 12.5 v/v% butanone	99.1% after 20000 cycles at 5 A g ⁻¹	This work
V ₂ O ₅ /graphene oxide	Zn foil	21 m LiTFSI + 3 m ZnOTf ₂ +10 wt %	93% over 600 cycles at 500	ACS Appl. Mater.

		PVA	mA g ⁻¹	Interfaces, 2020, 12, 15305
V ₂ O ₅	Zn foil	2 M ZnSO ₄ + 25% sulfolane	70% after 500 cycles at 10 A g ⁻¹	Angew. Chem. Int. Ed., 2023, 62, e202214966
V ₂ O ₅	Zn foil	Zn(OTf) ₂ + 30% 2- propanol	79.6% after 1500 cycles at 2 A g ⁻¹	Adv. Mater., 2022, 34, 2207344
V ₂ O ₅ ·nH ₂ O	Zn foil	2 M Zn(OTf) ₂ + 40% DME	93.1% after 2000 cycles at 2 A g ⁻¹	Energy Storage Mater., 2022, 47, 203
NVO	Zn foil	2 M ZnSO ₄ + 0.1 M ImS	88% after 3000 cycles at 20 A g ⁻¹	Energy Environ. Sci., 2022, 15, 4748
Zn _x V ₂ O ₅ ·nH ₂ O	Zn foil	3 M ZnSO ₄ + 68% EG	89.6% after 500 cycles at 500 mA g ⁻¹	Nano Energy, 2021, 80, 105478

Response to the comment “*More importantly, the explanations to why and how butanone protects cathode are not convincing and in fact contradictory to the experimental results. For example, the color change in both LCE and WSE to yellowish is a strong indicator of V-oxide dissolution, which is contradictory to the conclusion that butanone stops dissolution.*”:

As we discussed in the earlier version of response letter, the bottle containing WSE shows lighter yellow compared with that of LCE, indicating the dissolution of NVO is minished by butanone (Fig. R2a). We did not claim that butanone can fully stop the dissolution of NVO under that test condition, but it can greatly suppress the dissolution of NVO. Moreover, the above dissolution test is just a cursory evidence to prove the dissolution of NVO can be minished since the used electrolyte is

extremely overdosed for working condition of aqueous Zn ion battery. To avoid such misunderstandings, we reperformed the dissolution test in 2032-type coin cells, in which 80 μL electrolyte were used. After resting for 7 days, the separators of cells with LCE and WSE were taken out and further used for ICP test. As shown in Fig. R2b, the atom ratio of V element for LCE and WSE is 0.082 mg L^{-1} and 0.01 mg L^{-1} , indicating that WSE can highly suppress the dissolution of NVO under the actual working condition. As we also discussed in the previous response letter, to highlight the negative impact of cathode dissolution on the overall performance of Zn/NVO cells, NVO electrodes that underwent the dissolution tests (resting the Zn/NVO cells with LCE and WSE for 7 days) were subjected to GCD tests. Compared to the cells that did not experience the dissolution test (Fig. 3a), the NVO electrodes tested in LCE and WSE show capacity loss of 44.4 % and 7.8% in the first discharge process, respectively, indicating WSE can highly suppress the cathode dissolution of NVO (Fig. R2c).

Fig. R2 **a** Optical images of NVO electrode immersed in LCE and WSE for 7 days. **b** The dissolved V content after resting NVO in coin cell with LCE and WSE as electrolytes for 144 h. **c** The first discharge curves of Zn/NVO cells after resting for 144 h in LCE and WSE.

Response to the comment “*In V-chemistry, the appearance of low 2-theta (001) peak is an indicative of formation of hydrated compound with layered structure in which H_2O is situated; this structure is favorable to $\text{Zn}^{2+}/\text{H}^+$ storage. However, Fig.3g of TGA shows the opposite. In addition, the 2-theta values related to (001) plane for NVO cycled in both LCE and WSE is within the error bar, 12.1° vs. 12.0° , implying that there is virtually no impact from butanone on cathode protection.*”:

Indeed, we agree that the insertion of **a suitable amount** of H_2O molecules into the layers of V-based cathode is beneficial to the $\text{Zn}^{2+}/\text{H}^+$ insertion. However, previous works also confirmed that V-based cathode experiences a dramatic lattice variation **as overdosed water molecules** intercalate

into its layered spacing, which is a potential reason for the capacity fading (*Adv. Funct. Mater.*, 2022, 2111714; *ACS Energy Lett.*, 2018, 3, 1366 and *Adv. Sci.*, 2021, 8, 2102053). To be more specific, as proved by Li and his co-workers, the suitable addition of propylene carbonate (PC) to $\text{Zn}(\text{OTf})_2$ aqueous electrolyte can decrease the number of solvated H_2O molecules to reduce the co-inserted amount of H_2O during the discharge process of NVO, hence achieving stable electrochemical performance of Zn/NVO cells (Fig. R3). Moreover, as proved by the investigation of other kinds of metal ion batteries, e.g. Li ion batteries and K ion batteries, the overdosed insertion of solvent could also increase the strain of cathode and result in consequent crack formation (*Nat. Energy* 2022, 7, 484-494; *Angew Chem. Ed. Int.* 2022, 134, e202208291).

Fig. R3 Schematic illustrations of the structure evolutions of NVO during continuous Zn^{2+} insertion/extraction processes in P0, P20 and P50 electrolyte, respectively (*Adv. Funct. Mater.* 2022, 2111714). [D.-S. Liu, Y. Zhang, S. Liu, L. Wei, S. You, D. Chen, M. Ye, Y. Yang, X. Rui, Y. Qin, C. C. Li, Regulating the Electrolyte Solvation Structure Enables Ultralong Lifespan Vanadium-Based Cathodes with Excellent Low-Temperature Performance. *Adv. Funct. Mater.* 2022, 32, 2111714. <https://doi.org/10.1002/adfm.202111714>. Copyright © 2022 Wiley-VCH GmbH]

In our case, we also found that the co-inserted amount of H_2O into NVO in WSE is decreased to some extent, due to the reduced solvated H_2O in the primary solvation sheath of Zn^{2+} . Note that H_2O molecules and Na^+ are pre-intercalated between the V_3O_8 layers and act as pillars for NVO

cathode via a facile hydrothermal method. To highlight the difference of NVO cycled in LCE and WSE, we dissected the Zn/NVO cells with LCE and WSE after 50 cycles at a low current density of 0.1 A g^{-1} . Note that the test condition is different from the previous manuscript (5 A g^{-1}) considering the electrochemical process is more adequate under low current density. Moreover, we set the cutoff voltage of NVO with both electrolytes as 1.6 V (fully charged state) to investigate the inserted lattice H_2O into NVO in different electrolytes. Several characterizations including TG, XPS, XRD and SEM were performed to evaluate the structure evolution of different NVO electrodes. In TG curves, the weight loss above $200 \text{ }^\circ\text{C}$ could be attributed to the loss of lattice H_2O . Fig. R4a suggests the NVO cycled in WSE shows less loss of lattice H_2O , indicating the amount of co-inserted H_2O is decreased to some extent. O 1s XPS spectra further confirm the decreased H_2O of NVO with WSE, as the intensity of peak representing O-H at 532.5 eV of NVO with WSE is lower than that with LCE (Fig. R4b). As shown in Fig. R4c, the XRD of NVO cycled in LCE experiences a more obvious lattice expansion than that in WSE (11.3° to 11.1°), as revealed by the more deviated (001) plane from the pristine NVO. Such lattice expansion of NVO in LCE may causes strain and the consequent crack formation, as revealed by SEM images (Fig. R4d and e). Much more cracks were formed on the NVO electrode cycled in LCE while a flat surface is observed for NVO cycled in WSE, indicating protective ability of WSE for NVO at the electrode level. Based on the above experimental results, it can be concluded that H_2O co-insertion happens in both LCE and WSE, except more H_2O is co-inserted into NVO in LCE during the cycling process. Moreover, **the insertion of suitable amount of H_2O is beneficial for the energy storage process of NVO while excess insertion of H_2O can cause capacity fading.** Therefore, the protection mechanism of WSE for NVO can be concluded as: i) the excess co-insertion of H_2O is suppressed and the structure aberration/collapse of NVO during cycling is relieved; ii) the dissolution of NVO is greatly mitigated because the activity of H_2O is reduced owing to the intense hydrogen bonding interaction between H_2O and butanone.

Fig. 4 **a** TG curves, **b** O 1s XPS spectra, **c** XRD pattern, **d** and **e** SEM images of NVO electrode after 50 cycles at 0.1 A g^{-1} in LCE and WSE.

Response to the comment “Having said the above, it appears that the degradation shown in Fig. R8c of LCE full cell originates from the anode corrosion and cathode dissolution, and butanone’s role in WSE is to protect anode. Since lower concentration of ZnSO_4 has a higher pH, it will corrode the anode more severely. Fig.R8c indeed shows faster degradation and lower capacity at lower ZnSO_4 concentrations.”:

As we have demonstrated that the structure degradation and dissolution of NVO are also relieved in WSE in the above, the role of butanone in WSE is to simultaneously protect NVO cathode and Zn anode. Additionally, we agree that the corrosion of Zn anode is more severe under higher pH solution, but only for alkaline Zn-based battery. Since ZnSO_4 is a strong acid and weak base salt, the pH of ZnSO_4 solution increases as the concentration decreases, but it is always mildly acidic. Therefore, the corrosion of Zn anode is relieved in low-concentrated electrolyte with higher pH but always lower than $\text{pH}=7$, which is also reported by recently reported work (*Adv. Energy Mater.*, 2022, 12, 2102982). The reasons for fast capacity fading of Zn/NVO cell using electrolyte with lower ZnSO_4 concentration in Supplementary Fig. 2 could be multiple, including low ionic conductivity of electrolyte, high dissolution tendency of NVO and higher solvation degree of Zn^{2+} that causes overdosed H_2O co-insertion.

Other comments:

• *pH value for 0.5M ZnSO₄ is questionable. It appears much lower than the literature data as well this reviewer's own experience (pH~5.0), which prompts the question if other pH value is accurate, or pH meter is calibrated before each measurement??*

Response: Thanks for your comment. We retested the pH value and confirmed the pH value is accurate and the pH meter is calibrated before each measurement. We further prepared 2 M ZnSO₄ and compared its pH value with reported works since 2 M ZnSO₄ is more widely used as electrolyte. The pH value of our 2 M ZnSO₄ is 3.6, which is similar to some works such as *Small*, 2023, 19, 2302161 (3.6), *ChemElectroChem*, 2021, 8, 3553–3566 (3.6), *J. Mater. Chem. A*, 2022, 10, 6636–6640 (3.8) and *Adv. Energy Mater.*, 2021, 11, 2101158 (3.9).

• *There is no explanation on why adding butanone to ZnSO₄ reduces pH.*

Response: Thanks for your comment. LCE is prepared by dissolving 5.75 g of ZnSO₄·7H₂O into 40 mL of deionized water while WSE is prepared by dissolving 5.75 g of ZnSO₄·7H₂O into a mixed solution containing 5 mL of butanone and 35 mL of deionized water. Since butanone does not ionize in aqueous solution, butanone itself has no impact on the pH variation of WSE. However, it cannot dissolve ZnSO₄ and the introduction of butanone decreases the content of H₂O from 40 mL in LCE to 35 mL in WSE. Therefore, the H₂O in WSE would dissolve more ZnSO₄ salt compared to that in LCE. Considering ZnSO₄ is hydrolyzing in aqueous solution, more ZnSO₄ dissolved by H₂O in WSE will lead to more severe hydrolysis thus the pH of WSE is slightly lower than that of LCE. We added the related discussion in the revised manuscript.

• *Fig R11 (Fig.3i) only shows higher D at higher DoD, why? How accurate is GITT method at high DoD?*

Response: Thanks for your comment. The ion diffusion coefficient (D_{ion}) at low DOD reflects the H⁺ insertion kinetics while D_{ion} at high DOD represents the Zn²⁺ insertion kinetics (*J. Am. Chem. Soc.*, 2017, 139, 29, 9775–9778; *Angew. Chem. Int. Ed.*, 2022, 134, e202207779 and *Adv. Energy Mater.*, 2022, 12, 2201434). As shown in Fig. 3i, the D_{ion} of H⁺ insertion is similar in LCE and WSE, indicating the introduction of butanone has negligible effect on H⁺ insertion kinetics. However, the D_{ion} of Zn²⁺ insertion in WSE is higher than that in LCE, suggesting the faster Zn²⁺ insertion kinetics

with newly recruited butanone in WSE. This phenomenon is because that i) WSE shows boosted Zn^{2+} desolvation kinetics; ii) the overdosed co-inserted H_2O is minished in WSE. The excess insertion of H_2O in LCE can block the Zn^{2+} insertion pathway, thus increasing the insertion resistance. We have added the related discussion in the revised manuscript.

GITT method has been wildly used to investigate the ion insertion kinetics during the whole discharge process, not only for aqueous Zn ion batteries, but also for Li ion batteries and other ion batteries (*Adv. Mater.*, 2018, 30, 1705580; *J. Phys. Chem. C*, 2010, 114, 51, 22751–22757 and *Adv. Energy Mater.*, 2013, 3, 128–133). Therefore, the accuracy of it should not be worried. To dispel the misgivings of this reviewer, we have added detailed test parameters of GITT in the experimental section.

• *Fig.3c, cycle stability at 5A/g should be tested for more than thousands of hours data to be competitive with literature data that have shown systems better stability even without additive.*

Response: Thanks for your comment. Limited by the revision time, we further performed 20000-cycle long-term cycling test of Zn/NVO cells with LCE and WSE, which is among the longest cycle number of reported V-based aqueous Zn ion batteries. As shown in Fig. R5, Zn/NVO cell with WSE exhibits a capacity retention of 99.1% while that with LCE fails quickly after only 400 cycles. Additionally, the cycling stability of Zn/NVO cells with WSE is better than that of Zn/V cells based on other electrolyte chemistry or electrode modification, confirming the effectiveness of WSE. (Table R2).

Fig. R5 Cycling stability comparison at 5 A g⁻¹ of Zn/NVO cells in LCE and WSE.

Table R2. Comparison of the electrochemical stability of Zn-V battery with different electrolyte chemistry.

Cathode	Anode	Electrolyte	Capacity retention	Ref.
NVO	Zn foil	0.5 M ZnSO ₄ + 12.5 v/v% butanone	99.1% after 20000 cycles at 5 A g ⁻¹	This work
V ₂ O ₅ /graphene oxide	Zn foil	21 m LiTFSI + 3 m ZnOTf ₂ +10 wt % PVA	93% over 600 cycles at 500 mA g ⁻¹	ACS Appl. Mater. Interfaces, 2020, 12, 15305
V ₂ O ₅	Zn foil	2 M ZnSO ₄ + 25% sulfolane	70% after 500 cycles at 10 A g ⁻¹	Angew. Chem. Int. Ed., 2023, 62, e202214966
V ₂ O ₅	Zn foil	Zn(OTf) ₂ + 30% 2- propanol	79.6% after 1500 cycles at 2 A g ⁻¹	Adv. Mater., 2022, 34, 2207344
V ₂ O ₅ ·nH ₂ O	Zn foil	2 M Zn(OTf) ₂ + 40% DME	93.1% after 2000 cycles at 2 A g ⁻¹	Energy Storage Mater., 2022, 47, 203
NVO	Zn foil	2 M ZnSO ₄ + 0.1 M ImS	88% after 3000 cycles at 20 A g ⁻¹	Energy Environ. Sci., 2022, 15, 4748
Zn _x V ₂ O ₅ ·nH ₂ O	Zn foil	3 M ZnSO ₄ + 68% EG	89.6% after 500 cycles at 500 mA g ⁻¹	Nano Energy, 2021, 80, 105478

REVIEWERS' COMMENTS

Reviewer #2 (Remarks to the Author):

The authors have added new data to address this reviewer's concerns. These new data along with new discussion justify the acceptance of the paper in the current form.